# Gene editing for latent herpes simplex virus infection reduces viral load and shedding in vivo

Martine Aubert [1], Anoria K. Haick[1], Daniel E. Strongin[2], Lindsay M. Klouser [1,2], Michelle A. Loprieno [1], Laurence Stensland[2], Tracy K. Santo[2], Meei-Li Huang[2], Ollivier Hyrien[1], Daniel Stone [1] & Keith R. Jerome [1,2] ✉

Anti-HSV therapies are only suppressive because they do not eliminate latent HSV present in ganglionic neurons, the source of recurrent disease. We have developed a potentially curative approach against HSV infection, based on gene editing using HSV-specific meganucleases delivered by adeno-associated virus (AAV) vectors. Gene editing performed with two anti-HSV-1 meganucleases delivered by a combination of AAV9, AAV-Dj/8, and AAV-Rh10 can eliminate 90% or more of latent HSV DNA in mouse models of orofacial infection, and up to 97% of latent HSV DNA in mouse models of genital infection. Using a pharmacological approach to reactivate latent HSV-1, we demonstrate that ganglionic viral load reduction leads to a significant decrease of viral shedding in treated female mice. While therapy is well tolerated, in some instances, we observe hepatotoxicity at high doses and subtle histological evidence of neuronal injury without observable neurological signs or deficits. Simplification of the regimen through use of a single serotype (AAV9) delivering single meganuclease targeting a duplicated region of the HSV genome, dose reduction, and use of a neuron-specific promoter each results in improved tolerability while retaining efficacy. These results reinforce the curative potential of gene editing for HSV disease.

HSV infections can cause recurrent orofacial, corneal, anogenital, or other lesions, and infections of newborns can lead to disseminated disease and devastating neurological sequelae. Genital infection with HSV-2 increases the risk of acquisition of HIV, and is a major driver of the global HIV pandemic[1]. Current antiviral therapy can treat acute episodes and suppress outbreaks, but does not cure established infection[2–6]. Recurrent outbreaks result from the ability of HSV to establish latent infection within ganglionic neurons innervating the affected sites. Latent HSV in ganglia is unaffected by traditional antivirals, explaining the inability of antivirals to cure, and reactivations typically commence again once therapy is stopped.

A promising potentially curative strategy involves gene editing directed at latent HSV itself[7]. In a recent study, AAV-delivered meganucleases eliminated over 90% of HSV-1 genomes from the superior cervical ganglia of latently infected mice[8]. Despite this impressive reduction in ganglionic HSV loads after gene editing, the relevance that such reduction would have for human HSV infection is uncertain. Infected persons are typically not concerned with ganglionic viral loads per se, but instead about symptomatic disease and/or viral shedding, and the associated risk of transmission to others[9]. Existing mouse models are limited in their ability to address these issues, since latently infected mice rarely spontaneously reactivate HSV or show viral shedding at peripheral tissues.

[1]Vaccine and Infectious Disease Division, Fred Hutchinson Cancer Center, Seattle, WA 98109, USA. [2]Department of Laboratory Medicine and Pathology, University of Washington, Seattle, WA 98133, USA. ✉e-mail: kjerome@fredhutch.org

Here, we introduce a model of small-molecule induction of HSV-1 reactivation and peripheral shedding in latently infected mice, and demonstrate that gene editing mediates a dramatic reduction not only in ganglionic viral loads, but also in induced viral shedding. Optimization of the therapeutic approach through regimen simplification, dose reduction, and tissue restriction of meganuclease expression results in almost complete elimination of undesired effects on liver and ganglia, supporting the continued clinical development of this strategy.

## Results

### Meganuclease therapy reduces ganglionic viral load after ocular or genital HSV infection

We previously evaluated several AAV serotypes for delivery of meganucleases to latently infected mice, and found the best results with AAV-Rh10, followed by AAV8 and AAV1[8]. To further improve efficacy, we tested additional neurotropic AAV serotypes, including AAV7, AAV9, AAV-DJ, and AAV-DJ/8[10,11], for delivery of the anti-HSV1 meganuclease, m5, at a dose of $10^{12}$ AAV genomes (vg) per mouse (Fig. S1a), using our model of orofacial HSV disease. Both AAV9 and AAV-Dj/8 were superior to $10^{12}$ vg AAV-Rh10, the best of our previously used serotypes[8], showing HSV reductions in superior cervical ganglia (SCG) of 95% ($p = 10^{-5}$) and 90% ($p = 0.018$), respectively, relative to untreated controls (Fig. S1b), comparing favorably with the 65% reduction previously obtained with m5 alone delivered with AAV-Rh10 [8]. Similarly, AAV9 and AAV-Dj/8 showed better activity than AAV-Rh10 in trigeminal ganglia (TG), with HSV load reductions of 48% ($p = 0.07$) and 41% ($p = 0.5$), respectively (Fig. S1b), compared with our prior observation of no detectable reduction using AAV-Rh10 delivering m5[8]. The route of AAV administration (retro-orbital vein vs. intradermally into the whisker pad) did not have any detectable impact on either AAV transduction or gene editing efficiencies (Fig. S1e–g).

We previously demonstrated that gene editing of HSV could be increased by using combinations of AAV serotypes for meganuclease delivery, rather than a single AAV serotype, a finding we ascribed to the heterogeneity of neuronal subsets within HSV-infected ganglia[8]. We therefore evaluated gene editing with the anti-HSV1 meganuclease m5, which cleaves a sequence in the *UL19* gene coding for the major capsid protein VP5[12], when delivered using single AAV serotypes vs. combinations of AAV9, AAV-Dj/8, and AAV-Rh10 (Fig. S2a) administered as a total dose of $10^{12}$ vg per mouse. In agreement with our previous results, combinations of AAV serotypes led to robust HSV gene editing, with the triple combination of AAV9, AAV-Dj/8, and AAV-Rh10 showing especially strong reductions in HSV loads and mutagenesis of residual HSV across both SCG and TG (Fig. S2b–g).

While orofacial infections with HSV are extremely common, genital infections, which lead to latent infection of dorsal root ganglia (DRG), also represent a major cause of morbidity. We therefore established latent genital infections in mice by intravaginal inoculation with HSV-1 after treatment with Depo Provera, which synchronizes the estrus cycle and increases HSV infection[13]. Infected mice were treated with a total dose of $3 \times 10^{12}$ vg of the AAV9, AAV-Dj/8, and AAV-Rh10 combination delivering two HSV1-specific meganucleases simultaneously (m5 along with m8, which targets a sequence in the *UL30* gene coding for the catalytic subunit of the viral DNA polymerase[12]. In parallel, we tested the same AAV combination against latent orofacial HSV infection as described above (Fig. 1a, b). Remarkably, efficacy in the vaginal model of infection was the highest we have observed to date, with a 97.7% reduction in HSV viral load in DRG (Fig. 1c). This compared favorably with the orofacial infection group treated in parallel, in which (in agreement with our previous studies) we observed robust gene editing with significant reductions of ganglionic HSV loads of 89% in SCG and 61% in TG (Fig. 1d, e).

### Induction of HSV shedding using the BET bromodomain inhibitor JQ1

Mice generally show little if any spontaneous HSV reactivation, with minimal to no viral shedding at peripheral sites, limiting their utility in cure studies. The BET (Bromo and Extra-Terminal domain) bromodomain inhibitor JQ1 was reported to reactivate latent HSV in vitro in primary neuronal cultures, and HSV could be detected in the eyes of about one-third of latently infected mice treated with JQ1[14]. To evaluate the utility of JQ1 for our cure work, we extended these studies to determine the quantitative kinetics of HSV shedding after JQ1 therapy.

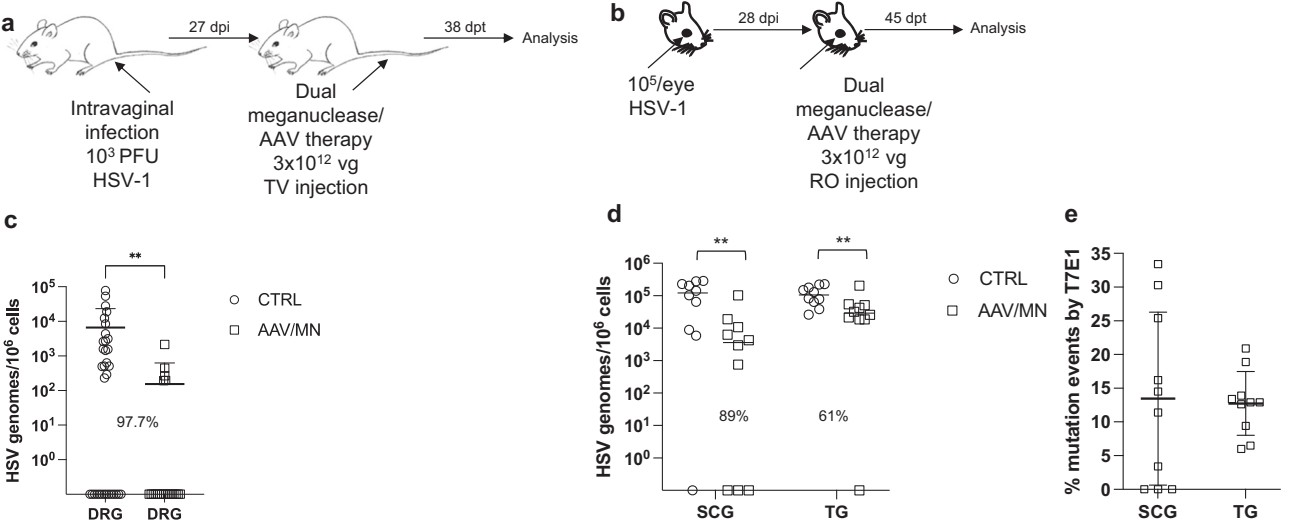

**Fig. 1 | Decrease of ganglionic HSV loads from genital and ocular infection after meganuclease therapy. a, b** Experimental timeline of (**a**) vaginal or (**b**) ocular infection and meganuclease therapy. RO, retroorbital; TV, tail vein. **c** HSV loads in DRGs from control ($n = 7$) and dual meganuclease-treated ($n = 4$) mice vaginally infected with HSV-1; $p = 0.001$. **d.** HSV loads in SCGs and TGs from control ($n = 10$) and dual meganuclease-treated ($n = 10$) mice ocularly infected with HSV-1; $p = 0.0046$ and 0.0034 for SCG and TG, respectively. **e** Gene editing at the m5 target site of residual virus quantified by T7E1 assay in SCG and TG from dual meganuclease-treated mice ($n = 10$). Each graph shows individual and mean values with standard deviation, percent decrease of HSV loads in treated mice compared to control mice and statistical analysis (unpaired one-tailed Mann-Whitney test with **$p < 0.01$). AAV loads are shown in Supplemental Fig. 9a, b. Source data are provided as a Source Data file.

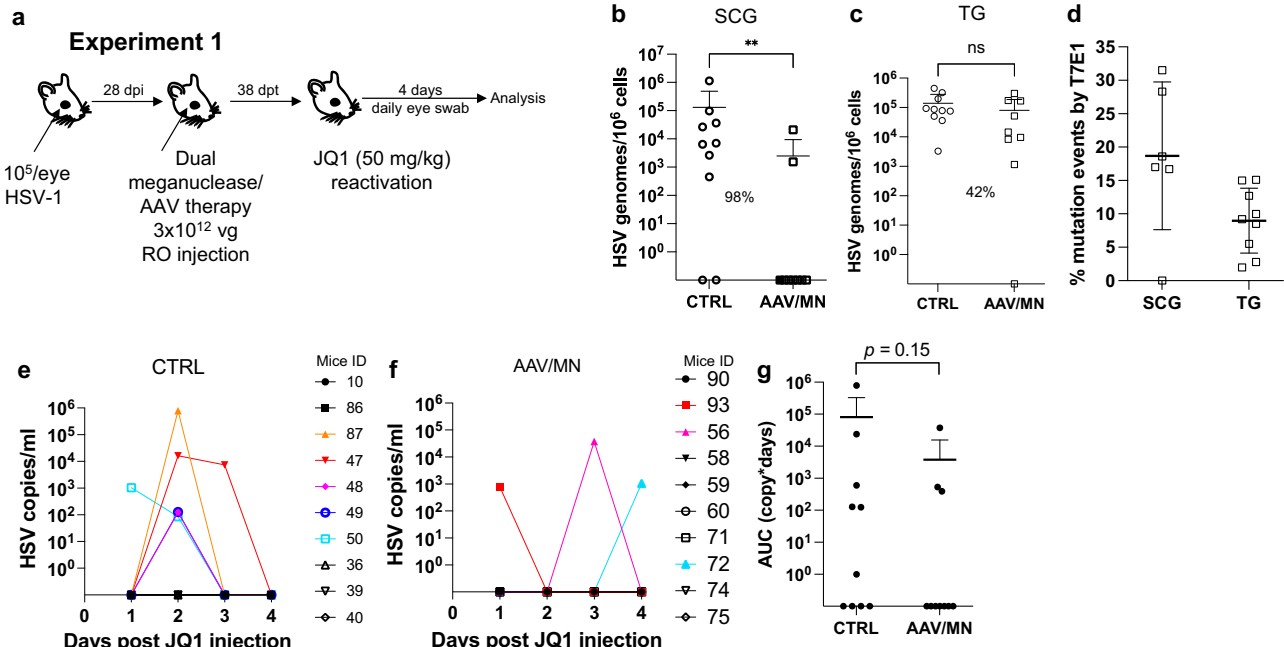

**Fig. 2 | Decrease of peripheral virus shedding in meganuclease-treated mice.** Experimental timeline of ocular infection, meganuclease treatment and viral reactivations with JQ1. **a** Experiment 1 (n = 10 per group). HSV loads in SCGs (**b**: p = 0.0057) and TGs (**c**). Percent decrease of HSV loads in treated mice compared to control mice and statistical analysis (unpaired one-tailed Mann–Whitney test with ns: not significant, **p < 0.01) are indicated. Gene editing at the m5 target site of residual virus quantified by T7E1 assay in SCG and TG from dual meganuclease-treated mice (**d**). HSV titers in eye swabs collected daily from day 1 to 4 post JQ1 reactivation from control (**e**) and dual meganuclease-treated (**f**) infected mice. Panels 2i-k show data for both SCG and both TG from each mouse. Area under the curve (AUC) analysis (**g**) with p value (unpaired one-tailed Mann–Whitney test). AAV loads are shown in Supplemental Fig. 9c, d. Each graph shows individual and mean values with standard deviation. Source data are provided as a Source Data file.

A single intraperitoneal (IP) injection of JQ1 (50 mg/kg) given to latently infected mice (Fig. S3a) led to detectable shedding from the eyes of 56% (5/9) of animals, compared with 0/9 animals treated with vehicle alone (Fig. S3b, c). Viral shedding was transient, peaking at 2 days post JQ1, with maximal viral loads ranging from about $10^2$ to almost $10^6$ copies/swab (Fig. S3c). A direct comparison suggested that JQ1 may be a more powerful reactivation stimulus for HSV than hyperthermic stress (HS)[15], which in our hands led to detectable virus shedding in less than 20% of animals (2/12 HS vs 4/10 JQ1), with peak shedding viral loads two logs lower than after JQ1 treatment (Fig. S3d–f). Sequential treatment with JQ1 at one-week intervals led to repeated shedding episodes with similar kinetics as observed above (Fig. S4a–e). Over the course of three sequential JQ1 reactivations (Fig. S4e), shedding from individual mice was stochastic; 10/12 (83%) mice shed detectable virus at least once, but only 1/12 (8%) shed after all three treatments, while 4/12 (33%) and 5/12 (42%) shed only after two or one of the three treatments, respectively (Fig. S4e and Table S1). Shedding was typically unilateral (only detected in one eye), despite the initial inoculation being to both eyes (unilateral shedding was observed in 33/37 (89%) of events). The side of shedding in one episode was not predictive of the side of future shedding events (Table S1). Importantly for cure studies, repeated weekly reactivation of virus with JQ1 up to 7 weekly injections did not change ganglionic viral loads compared with control animals (Fig. S4g, h).

### Reduction of ganglionic HSV load is associated with reduced peripheral shedding

The ability to reproducibly induce HSV reactivation and shedding with JQ1 allowed us to investigate the relationship between ganglionic viral load reduction using meganucleases and subsequent viral shedding at peripheral sites. Latently infected mice were treated as above using the AAV9, AAV-Dj/8, and AAV-Rh10 combination delivering HSV1-specific meganucleases m5 and m8 at a total dose of $3 \times 10^{12}$ vg, or left

untreated as controls. One month later, mice were administered JQ1, and eye swabs were collected daily for 4 days (Fig. 2a). Consistent with our previous results, ganglionic tissues from treated mice showed a 98% and 42% reduction in mean viral loads in SCG and TG, respectively, when compared to control untreated animals (Fig. 2b, c) and gene editing in the remaining viral genomes (Fig. 2d). After JQ1 administration, only 3/10 (30%) of the dual meganuclease-treated mice had detectable virus in eye swabs, compared with 5/10 (50%) of control untreated animals (Fig. 2e, f). The mean titer of HSV in positive eye swabs was $3 \times 10^4$ copies/ml in the meganuclease-treated animals, compared with $1.2 \times 10^5$ copies/ml in the control animals. Area under the curve analysis (AUC) demonstrated a 95% reduction (p = 0.15) in total viral shedding in treated vs. control animals (Fig. 2g). In a separate experiment performed similarly (Fig. 3a), ganglionic tissues from treated mice showed 97% reduction in mean latent HSV genomes in both SCG and TG when compared to control untreated animals (Fig. 3b, c) and gene editing in the remaining genomes (Fig. 3d). While 3/8 mice from the control group shed virus with a mean viral titer of $8.2 \times 10^5$ copies/ml, 0/8 meganuclease-treated mice had detectable shedding, representing a 100% decrease in total virus shed (p = 0.10, Fig. 3e–g).

### Safety of AAV/meganuclease therapy

AAV-vectored therapies are generally considered safe. Nevertheless, dose-limiting liver toxicity has been observed after AAV administration in humans, non-human primates, and mice, typically at doses of $2 \times 10^{14}$ vg/kg or greater. The $3 \times 10^{12}$ vg/animal dose (-1 × $10^{14}$ vg/kg) used in the experiments described in Figs. 1–3 approached the level associated with liver toxicity in previous studies. Across multiple studies we observed that 7/70 (10%) animals treated with the $3 \times 10^{12}$ vg/animal dose exhibited clinical signs consistent with hepatotoxicity, including weight change, bloating, and general health decline. Hepatotoxicity was confirmed in these animals by subsequent histopathological

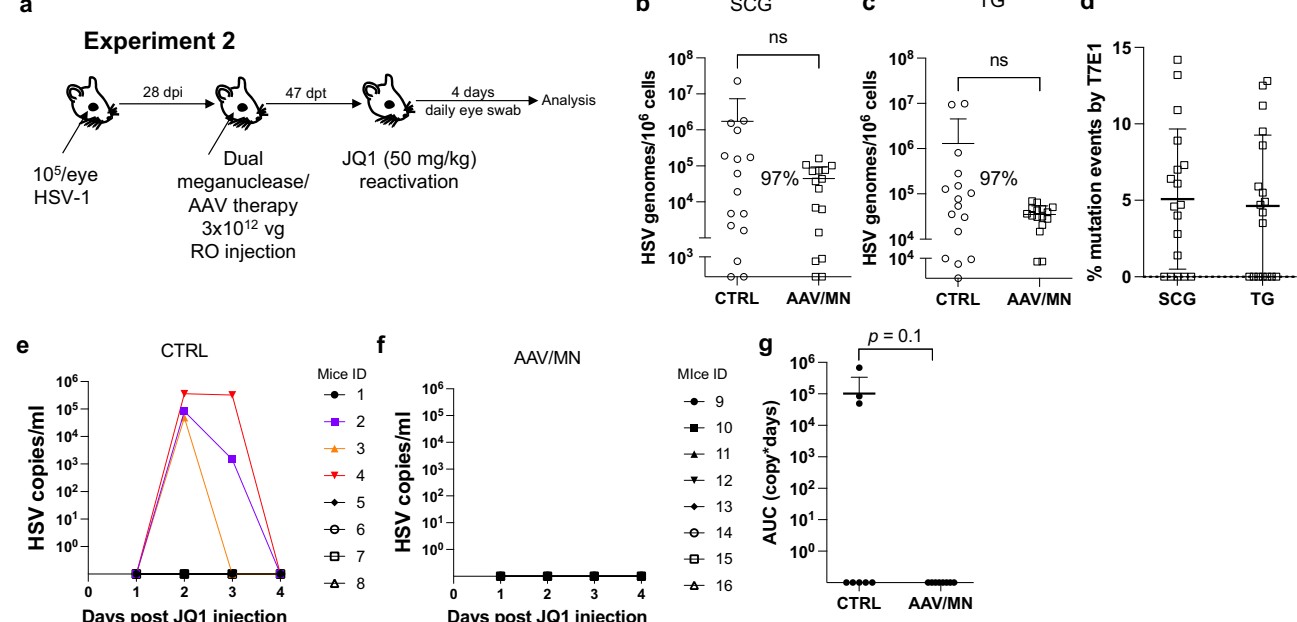

**Fig. 3 | Decrease of peripheral virus shedding in meganuclease-treated mice.**
Experimental timeline of ocular infection, meganuclease treatment and viral reactivations with JQ1. **a** Experiment 2 ($n = 8$ per group). HSV loads in SCGs (**b**) and TGs (**c**). Percent decrease of HSV loads in treated mice compared to control mice and statistical analysis (unpaired one-tailed Mann-Whitney test with ns: not significant, **$p < 0.01$) are indicated. Gene editing at the m5 target site of residual virus quantified by T7E1 assay in SCG and TG from dual meganuclease-treated mice (**d**). HSV titers in eye swabs collected daily from day 1 to 4 post JQ1 reactivation from control (**e**) and dual meganuclease-treated (**f**) infected mice. Panels 2**b-d** show data for both SCG and both TG from each mouse. Area under the curve (AUC) analysis (**g**) with $p$ value (unpaired one-tailed Mann–Whitney test). AAV loads are shown in Supplemental Fig. 9e, f. Each graph shows individual and mean values with standard deviation. Source data are provided as a Source Data file.

evaluation (Fig. S5 and Table S2). We therefore evaluated lower total doses of triple AAV serotype/dual meganuclease therapy (0.6, 1.2, or $1.8 \times 10^{12}$ vg/animal or 1.8, 3.6, or $5.4 \times 10^{13}$ vg/kg) for their tolerability and effects on viral load and JQ1-induced HSV shedding (Fig. 4a). These doses showed substantially improved tolerability, both clinically and upon histopathological examination and quantification of the number of inflammatory cell foci (ICF) in livers (Fig. S6a). Dose-dependent reductions in ganglionic HSV loads were observed across the three treatment groups compared to controls, ranging from 69% and 47% in SCG and TG, respectively, at the $0.6 \times 10^{12}$ dose to 94% and 73% at the $1.8 \times 10^{12}$ dose (Fig. 4b, c). To evaluate the effect of these reduced doses on HSV shedding, treated mice were subjected to three weekly rounds of JQ1 administration, followed by eye swabbing as described above. While the percentage of dual meganuclease-treated animals shedding virus after the first JQ1 reactivation was not reduced compared with the control mice, it was substantially lower than controls at all doses by the third JQ1 reactivation (0% (0/12), 8% (1/12) and 0% (0/12) for 0.6, 1.2, and $1.8 \times 10^{12}$ vg/mouse groups, respectively, versus 18% (2/11) in the control group) (Fig. 4f–i). This finding that may relate to the two additional weeks available for meganuclease expression and gene editing activity by the time of the third JQ1 reactivation. Consistent with this interpretation, the reduction in total viral shedding, as determined by AUC analysis, appeared to become more complete over time, with up to a 97–100% reduction in all three groups by the final JQ1 reactivation (Fig. 4j–l). The efficacy of reduced-dose dual meganuclease therapy ($1.8 \times 10^{12}$ vg) was confirmed in a separate experiment (Fig. 5a), showing a significant decrease in ganglionic viral loads in both SCG and TG (Fig. 5b, c). In this experiment, 7 of 12 control animals showed detectable viral shedding after JQ1 reactivation, compared with only 1 of 12 animals treated with AAV-meganuclease therapy, (Fig. 5d, e) and reduction in total viral shedding, as determined by AUC analysis (Fig. 5f, g). While none of the treated mice exhibited any clinical signs of hepatotoxicity, we did observe higher numbers of ICF

in liver of treated animals receiving the $1.8 \times 10^{12}$ vg dose compared to control mice (Fig. S6b). Histologic analysis of H&E stained TG sections from both control and treated animals revealed subtle evidence of neuronal injury, manifesting as neuronal degeneration, necrosis, and axonopathy. The scores grading prevalence and severity of the microscopic changes were higher in treated animals compared to control mice (Fig. S7 and Table S3). However, no mice in either the control or experimental group showed detectable signs of neuropathy.

### Reduced dose of AAV/meganuclease treatment of genitally infected animals decreases DRG latent viral load and may reduce genital HSV shedding

As noted above, genital HSV infection is a major cause of morbidity in humans. We therefore evaluated the reduced-dose dual meganuclease therapy (total dose of $1.8 \times 10^{12}$ vg/animal) in vaginally infected mice (Fig. 6a). In agreement with our previous results, the reduced-dose therapy led to a 78.8% ($p = 0.02$) to 95.6% ($p = 0.006$) reduction in latent virus genomes in DRGs (Fig. 6b).

We then sought to evaluate whether JQ1 could induce HSV shedding in the genital infection model, as we previously observed in the ocular infection model. Over 3 sequential JQ1 reactivations, only 2 of 8 control animals (and 1 of 8 AAV/meganuclease-treated animals) shed detectable virus, a rate lower than the 40–50% reactivation we typically observe after ocular infection (Fig. 6c, d). The apparently lower rate of reactivation seen in the vaginal model compared to the ocular model may be due to lower levels of ganglionic HSV loads in the DRG ($10^2$–$10^3$ vg/$10^6$ cells in DRG, Fig. 6b vs $10^4$–$10^5$ vg/$10^6$ cells in SCG or TG, Fig. 5b, c). While this lower reactivation rate prohibited meaningful statistical analysis, the observation that 2 out of the 8 control mice shed virus over 2 to 3 sequential days, while only 1 of the 8 AAV-treated mice shed virus, on a single day and at a substantially lower level, is qualitatively in agreement with our observations after ocular infection (Fig. 6c–g).

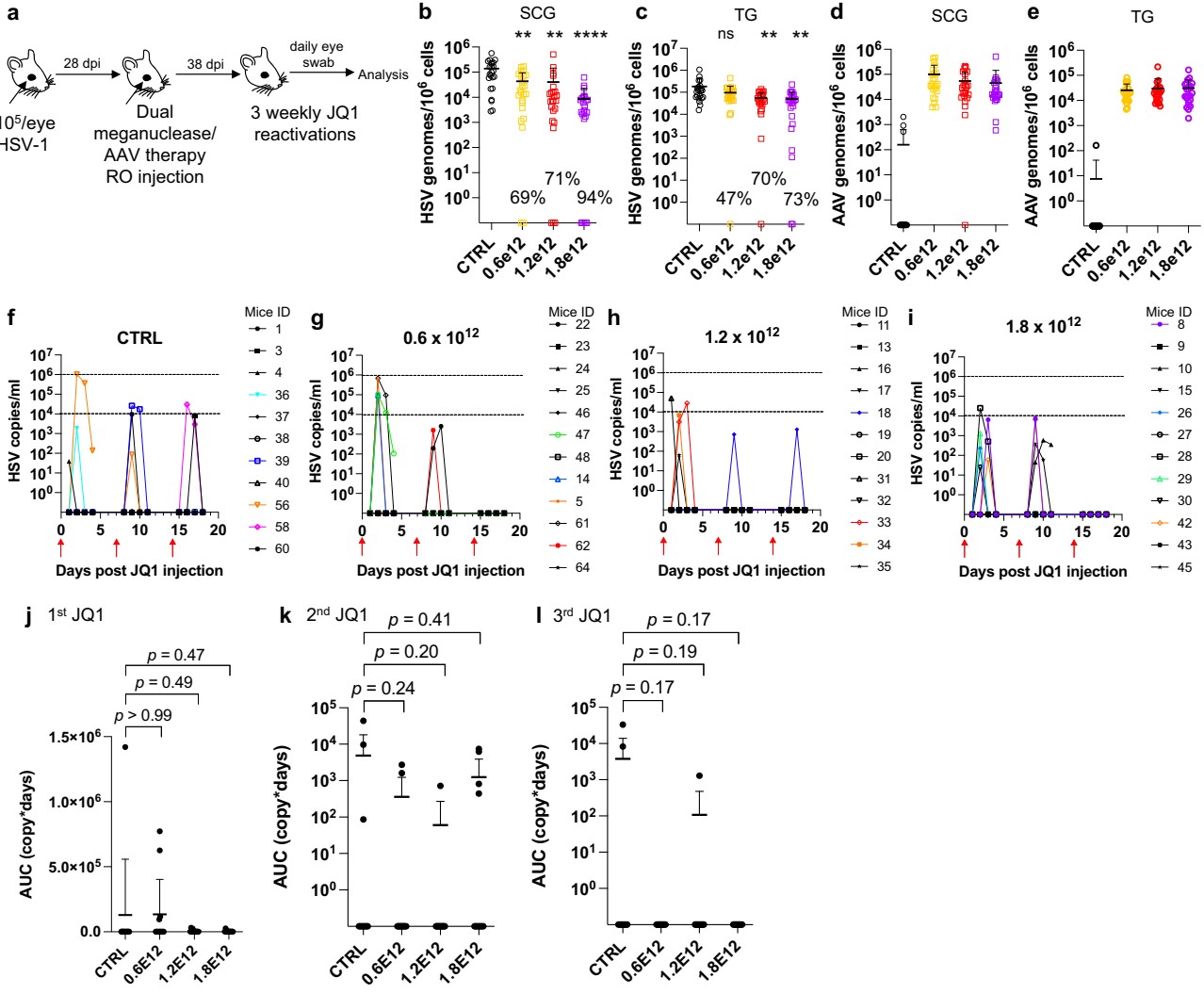

**Fig. 4 | Decrease of peripheral virus shedding in meganuclease-treated mice.** **a** Experimental timeline of ocular infection, meganuclease treatment and viral reactivations with JQ1. **b, c** HSV loads and **d, e,** AAV loads in SCGs (**b**; $p = 0.0016$, 0.0012, and <0.0001 for 0.6, 1.2, and 1.8 × $10^{12}$, respectively, **d**) and TGs (**c**; $p = 0.068$, 0.0025 and 0.0016 for 0.6, 1.2, and 1.8 × $10^{12}$, respectively, **e**) in control infected mice ($n = 11$) and infected mice treated with dual therapy delivered with 0.6 ($n = 12$), 1.2 ($n = 12$), or 1.8 ($n = 12$) × $10^{12}$ total vg AAV dose. Percent decrease of HSV loads in treated mice compared to control mice and statistical analysis (ordinary one-way Anova, multiple comparisons with ns: not significant, **$p < 0.01$, ****$p < 0.0001$) are indicated. **f–i** Virus titers in eye swabs collected daily from day 1 to 4 after each weekly JQ1 reactivation from control infected mice (**f**) and infected mice treated with dual therapy delivered with 0.6 (**g**), 1.2 (**h**), or 1.8 (**i**) × $10^{12}$ total vg AAV dose. **j–l** Area under the curve (AUC) analysis of virus shedding after first (**j**), second (**k**), and third (**l**) JQ1 reactivation from control infected mice ($n = 11$) and infected mice treated with dual therapy delivered with 0.6 ($n = 12$), 1.2 ($n = 12$) or 1.8 ($n = 12$) × $10^{12}$ total vg AAV dose. $p$ values (unpaired, ordinary one-way Anova, with multiple comparisons) compared virus shedding between treatment groups and the control group. Each graph shows individual and mean values with standard deviation. Source data are provided as a Source Data file.

## Meta-analysis of the effect of AAV/meganuclease therapy on HSV shedding

The stochastic nature of clinical HSV reactivation[16], recapitulated when induced by JQ1 in mice (Fig. S4 and Table S1), makes evaluation of viral shedding extremely resource-intensive. Practical constraints, including the number of animals that can be housed and studied at the same time, along with the extended duration of each study (~3 months), limited the statistical power of our individual experiments. We therefore performed a meta-analysis of data from all experiments presented above (Figs. 1–6), combining evidence from infection sites (orofacial or genital), thus comparing 174 swabs from AAV/meganuclease-treated animals to 99 swabs from experimentally-matched controls. The primary endpoint was viral shedding, expressed either as a binary variable (equal to 1 for samples in which HSV was detected and 0 otherwise) or the log10-transformed AUC for quantitative viral shedding. The experiments depicted in Figs. 1–6 represent all of the shedding studies

with the dual meganuclease/triple AAV therapy we have performed as of this writing, and each suggests a strong and consistent trend toward a substantial reduction in viral shedding after AAV/meganuclease therapy. Across all studies, the proportion of swabs with detectable HSV was 48% lower among AAV/meganuclease-treated animals compared to controls. The meta-analysis confirmed that animals receiving AAV/meganuclease therapy had a statistically significant reduction in viral shedding (OR = 0.41, $p = 0.010$, by generalized linear mixed models, GLMM).

We then asked whether dose or duration of meganuclease therapy was associated with the probability of viral shedding (expressed as a binary variable) or the quantity of viral shedding (expressed as the log10-transformed AUC). Overall, the probability of viral shedding significantly decreased with the dose of AAV/meganuclease (OR = 0.66; $p = 0.023$, GLMM), and also with the duration of meganuclease therapy (OR = 0.42; $p < 0.001$, GLMM) in treated animals compared to

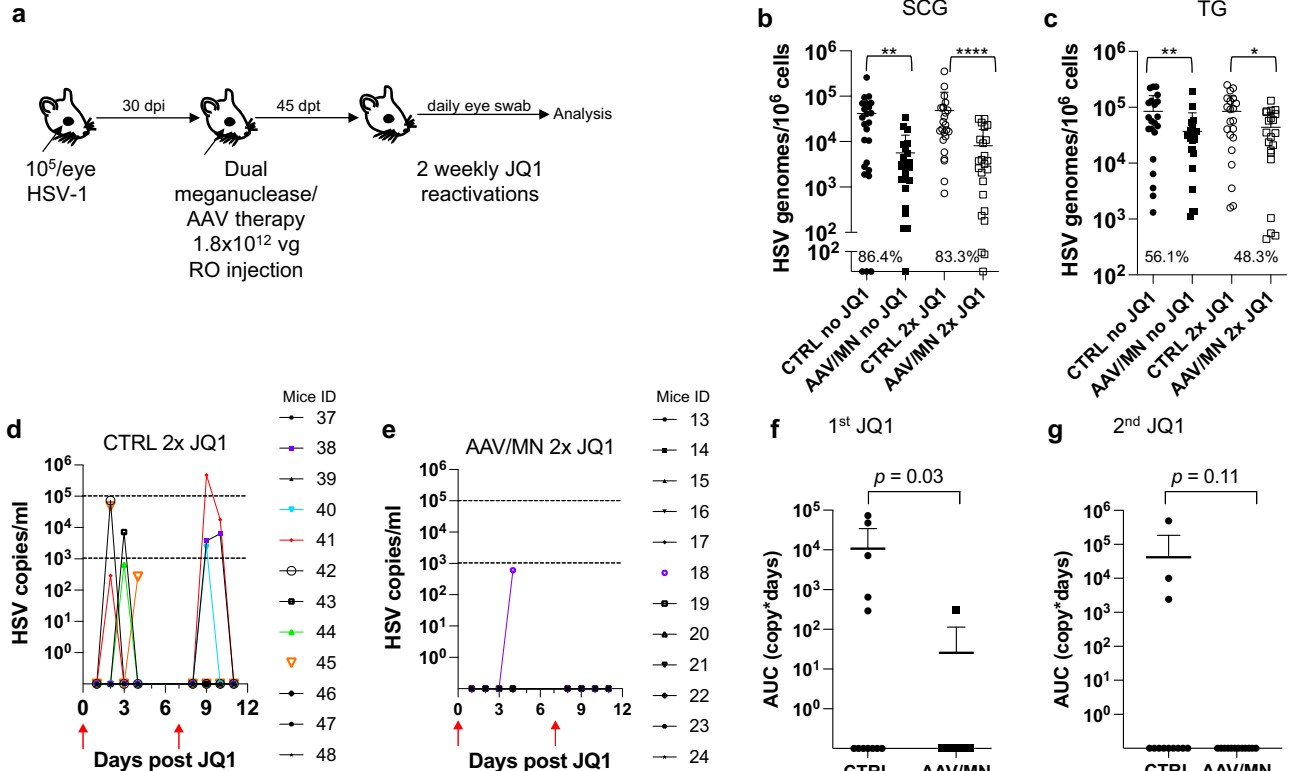

**Fig. 5 | Peripheral virus shedding decreases in dual meganuclease-treated mice.**
**a** Experimental timeline of ocular infection, meganuclease treatment and viral reactivations with JQ1. **b** HSV loads in SCGs (**b**; $p = 0.0012$, and $<0.0001$ for CTRL vs AAV/MN no JQ1 and for CTRL vs AAV/MN 2x JQ1, respectively), and TGs (**c**; $p = 0.0089$, and $0.0293$ for CTRL vs AAV/MN no JQ1 and for CTRL vs AAV/MN 2x JQ1, respectively) of control infected mice either unreactivated (CTRL no JQ1) or reactivated (CTRL 2x JQ1) and infected mice treated with dual therapy delivered with $1.8 \times 10^{12}$ total AAV dose either unreactivated (AAV/MN no JQ1) or reactivated (AAV/MN 2x JQ1), with $n = 12$ per group. Percent decrease of HSV loads in treated mice compared to control mice and statistical analysis (unpaired one-tailed Mann–Whitney test with *$p < 0.05$; **$p < 0.01$, ****$p < 0.0001$) are indicated. **d**, **e** Virus titers

in eye swabs collected daily from day 1 to 4 after two weekly JQ1 reactivations (red arrows) from control infected mice (**d**) and infected mice treated with dual therapy delivered with $1.8 \times 10^{12}$ total AAV dose (**e**). **f**, **g** Area under the curve (AUC) analysis after the first (**f**), and the second (**g**) JQ1 reactivation from control infected mice either unreactivated (CTRL no JQ1) or reactivated (CTRL 2x JQ1) and infected mice treated with dual therapy delivered with $1.8 \times 10^{12}$ total AAV dose either unreactivated (AAV/MN no JQ1) or reactivated (AAV/MN 2x JQ1), with $n = 12$ per group. $p$ values (unpaired one-tailed Mann–Whitney test) are indicated. Each graph shows individual and mean values with standard deviation. AAV loads are shown in Supplemental Fig. 9g–h. Source data are provided as a Source Data file.

controls. The data further indicate that overall, the quantity of virus shed (AUC) significantly decreased with the AAV/meganuclease dose at a rate of -0.36 $\log_{10}$ copy-days per $10^{12}$ increase in dose (LMM; $p = 0.028$), and also with the duration of meganuclease therapy, at a rate of $-0.48$ $\log_{10}$ copy-days per additional week after treatment (LMM; $p = 0.017$). No significant association was detected between the log10-transformed AUC and the interaction between time and dose (LMM; $p = 0.59$).

**Simplification of the AAV-meganuclease regimen**
The studies described above were performed using a triple AAV serotype/dual meganuclease approach, resulting in each animal receiving a total of 6 unique vectors (3 serotypes × 2 meganucleases). Clinical translation of such a complex regimen could raise manufacturing and quality control issues. We therefore sought to simplify AAV/meganuclease therapy to reduce the complexity of our therapeutic regimen. We took advantage of the dual cutting meganuclease m4, which recognizes a sequence in the duplicated gene *ICP0* in the HSV-1 genome and was previously shown to induce significant decrease of latent viral loads in ganglia of latently infected mice[8]. Latently infected mice were administered a total dose of $5 \times 10^{11}$ vg of either the combination of AAV9, AAV-Dj/8, and AAV-Rh10, or each single AAV serotype delivering the HSV1-specific meganuclease m4 (Fig. 7a). Consistent with the results using the lower dose of $6 \times 10^{11}$ of the triple AAV-dual MN

therapy (Fig. 4), ganglionic tissues from treated mice with the triple AAV combination delivering m4 showed a 73.9% ($p < 0.0001$) and 43.7% ($p = 0.014$) reduction in mean viral loads in SCG and TG, respectively, when compared to control untreated animals. When m4 was delivered using single AAV serotypes, the data confirmed that AAV9 on its own could recapitulate the viral load decrease seen with the triple AAV serotype combination, with 77.8% ($p < 0.0001$) and 49% ($p = 0.0046$) reduction in mean viral loads in SCG and TG, respectively (Fig. 7b, c). Furthermore, mice having received AAV9 alone showed the lowest levels of liver inflammation of any of the groups, similar to those in the control liver (Fig. 7d). At this reduced dose, regardless of the AAV serotype combination used, no detectable neurotoxicity was observed compared to the control animals (Fig. 7e, f).

To confirm that a simplified regimen composed of AAV9-m4 was also able to reduce peripheral virus shedding, latently infected mice were treated as above using AAV9 delivering either HSV1-specific meganuclease m4 or a catalytically inactive version (m4i) at a dose of $1 \times 10^{12}$ vg. One month later, mice were subjected to two weekly rounds of JQ1 administration, followed by daily eye swabbing for 3 days (Fig. 8a, b). A decrease of ganglionic viral loads of 89.6% ($p < 0.0001$) and 69% ($p = 0.03$) in SCG and TG respectively, was observed in m4-treated mice but not in mice treated with the inactive form of the meganuclease m4i (Fig. 8c, d). Furthermore, 6 out of 9 control mice and 6 out of 10 m4i-treated mice shed virus after JQ1 reactivations,

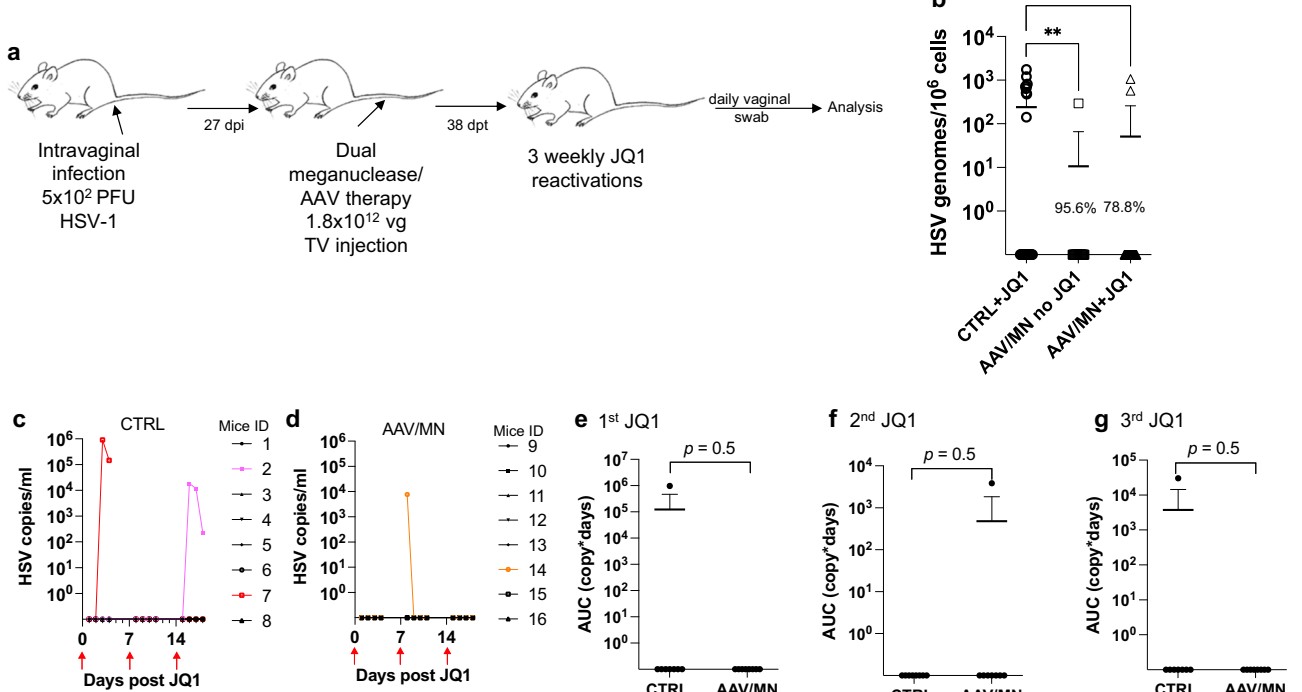

**Fig. 6 | Peripheral virus shedding decreases in dual meganuclease-treated mice.**
**a** Experimental timeline of intravaginal HSV-1 infection, meganuclease treatment and viral reactivations with JQ1. **b** HSV loads in DRGs from control infected mice reactivated with 3 weekly JQ1 injections and infected mice treated with dual therapy unreactivated, or reactivated with 3 weekly JQ1 injections with $n = 8$ per group; $p = 0.0055$, and 0.0198 (ordinary one-way Anova, multiple comparisons) for CTRL + JQ1 vs AAV/MN no JQ1 and for CTRL + JQ1 vs AAV/MN + JQ1, respectively. **c**, **d** HSV titers in vaginal swabs collected daily from day 1 to 4 post JQ1 injections

(red arrows) from control (**c**) and dual meganuclease-treated (**d**) infected mice. Area under the curve (AUC) analysis after the first (**e**), second (**f**), and third (**g**) JQ1 reactivation from control infected mice ($n = 8$) and infected mice treated with dual therapy, both reactivated with 3 weekly JQ1 injections ($n = 8$). $p$ values (unpaired one-tailed Mann–Whitney test) are indicated. Each graph shows individual and mean values with standard deviation. The AAV viral loads are shown in Supplemental Fig. 9i. Source data are provided as a Source Data file.

---

while only 3 out of 10 m4-treated mice had detectable virus shedding after reactivation (Fig. 8e–i). These data demonstrate that our simplified regimen can substantially reduce ganglionic viral loads, with an associated decrease in virus shedding after reactivation, and that these effects are dependent on an active enzyme and not on AAV itself. In this experiment, mice treated with m4 had slightly higher levels of liver ICF and TG axonopathy, but not more TG inflammation, compared to control mice (Fig. 8j–l).

### Tissue restriction of meganuclease expression improves tolerability

Across our studies, we observed that ~10% of the animals treated with a high dose of AAV/meganuclease ($2–3 \times 10^{12}$ vg/animal, or approximately $6–9 \times 10^{13}$ vg/kg) exhibited clinical signs consistent with hepatotoxicity, including weight change, bloating, and general health decline. When lower doses were evaluated, we observed substantially improved tolerability, both clinically and upon histopathological examination and ICF quantification. To further reduce hepatotoxicity, we evaluated the use of neuron-specific promoters (Calmodulin Kinase II (CamKII) and human Synapsin (hSyn)) combined with the CMV enhancer[17], to test the hypothesis that limiting enzyme expression to neuronal tissues would decrease or perhaps prevent liver toxicity (Fig. 9a). We found that latently infected mice treated with a high dose ($2 \times 10^{12}$ vg) of AAV9-E/CamKII-m4 or AAV9-E/hSyn-m4 did not show any clinical signs of hepatotoxicity (weight change, general health decline, or ICF), in contrast to mice treated with $2 \times 10^{12}$ vg AAV9-CBh-m4 (Fig. 9b, c). Moreover, while liver inflammation increased over time in AAV9-CBh-m4-treated mice, it remained low in AAV9-E/CamKII-m4 (Fig. S11d). Intriguingly, histopathologic signs of neurotoxicity in TG

from AAV9-E/CamKII-m4 or AAV9-E/hSyn-m4 treated mice were also similar to those in control mice, while they were significantly higher in TG from AAV9-CBh-m4 treated mice (Fig. 9d, e). A decrease of ganglionic viral loads of 67.9% ($p = 0.07$) and 70.4% ($p = 0.05$) in SCG and TG respectively, was observed in AAV9-E/CamKII-m4-treated mice but not in mice treated with the AAV9-E/hSyn-m4 (Fig. 9f, g). Assessment of m4 expression in neuronal tissues at different times post administration of either AAV9-CBh-m4 or AAV9-E/CamKII-m4 showed that the m4 expression increased over time but was in general slightly lower in tissues from AAV9-E/CamKII-m4-treated mice compared with AAV9-CBh-m4-treated mice (Fig. S11a, b). This may explain the slightly lesser degree of viral load reduction in AAV9- E/CamKII-m4-treated mice compared with AAV9- CBh-m4-treated mice (Fig. 9f, g). We conclude that the AAV9-E/CamKII-m4 regimen retains efficacy and shows improved tolerability compared to AAV9-CBh-m4.

## Discussion

Human infection with HSV is lifelong, and while current antiviral approaches can reduce symptoms and transmission, they do not cure. As such, there is a strong unmet desire for new and potentially curative approaches for HSV[9]. Here we extend our previous study of gene editing as a potential curative therapy for HSV in three important ways. First, we established a model of HSV reactivation in mice using a small molecule, to show that a reduction in ganglionic HSV loads via gene editing results in a significant reduction in viral shedding from mice with established orofacial infection. Second, we demonstrated high efficacy of gene editing of latent HSV in DRG after genital HSV infection. Third, we reduced or even eliminated the hepato- and neurotoxicity detected in some animals by decreasing the AAV dose,

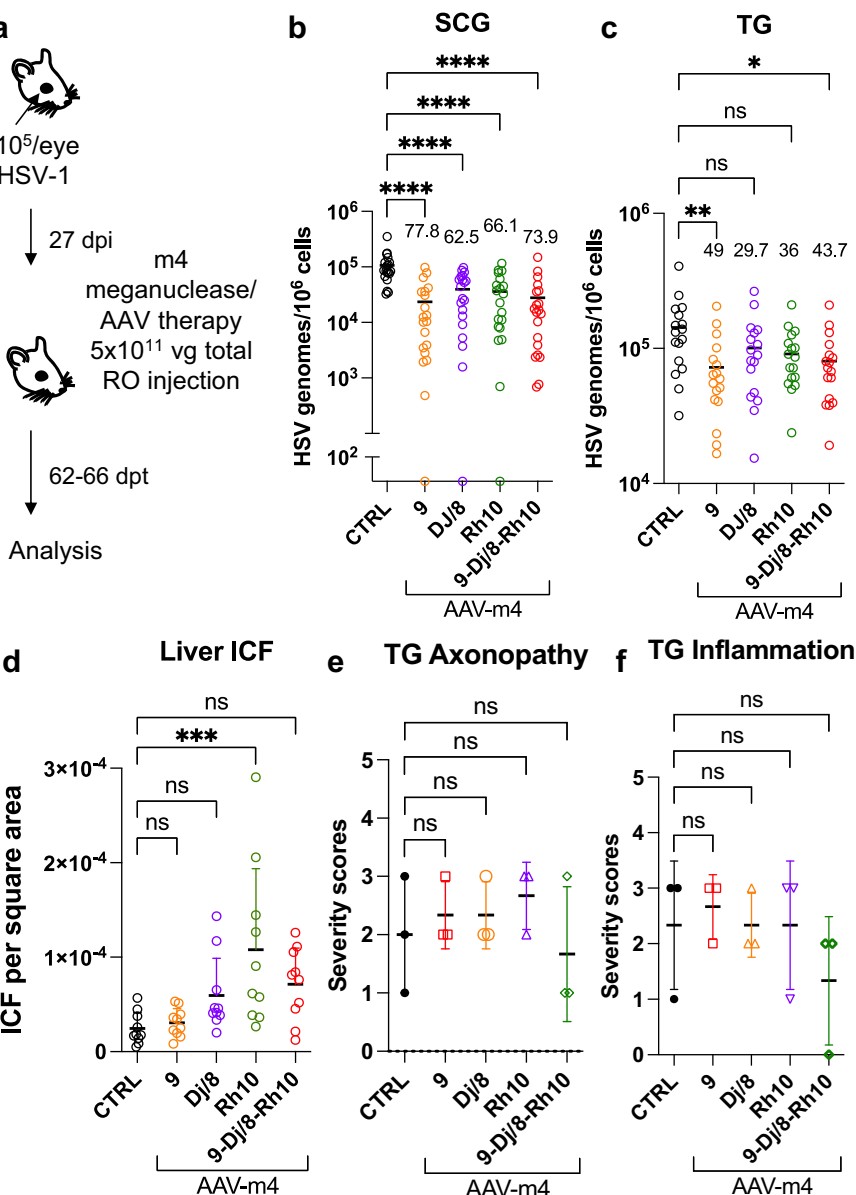

**Fig. 7 | Simplification of the meganuclease/AAV regimen. a** Experimental time-line of ocular infection and meganuclease therapy. **b**, **c** HSV loads in SCGs (**b**; p < 0.0001) and TGs (**c**; p = 0.0046 for AAV9 and 0.0142 for 9-Dj/8-Rh10) from infected control and infected mice treated with m4 delivered by retro-orbital (RO)) injections of $5 \times 10^{11}$ vg total of the single or triple combinations of AAV9, -Dj/8 and -Rh10. Percent decrease of HSV loads in treated mice (n = 10 per group) compared to control mice (n = 10) and statistical analysis (Ordinary one-way Anova, multiple comparisons with *p < 0.05; **p < 0.01, ****p < 0.0001; ns: not significant). **d** Inflammatory cell foci (ICF) in liver sections from either HSV-infected control

mice (n = 10), or mice treated with m4 delivered using AAV single or triple combinations of AAV9, -Dj/8, and -Rh10 (n = 10 per group); p = 0.0009 for Rh10. **e**, **f** Severity scores of axonopathy (**e**) and inflammation (**f**) in TG from infected control mice (n = 3 TG) and infected mice treated with m4 delivered using single or triple combinations of AAV9, -Dj/8 and -Rh10 (n = 3 TG per group) and statistical analysis (Ordinary one-way Anova, multiple comparisons with ns: not significant; ***p < 0.001). Each graph shows individual and mean values with standard deviation. The AAV viral loads are shown in Supplemental Fig. 9j–l. Source data are provided as a Source Data file.

simplifying the therapy regimen, and using a cell type-specific promoter. Together, these findings address several of the major drivers of interest in HSV cure[9], and support further development of gene editing for HSV infection.

Mice are easily infected with HSV, and have been critical in defining many aspects of HSV infection, latency, and immune control[18]. A major drawback, however, has been the fact that latent HSV infection in mice exhibits minimal to no spontaneous reactivation or peripheral virus shedding[19]. Thus, mice have been of limited utility in studying HSV therapeutics or vaccines that are directed at control of latent infection and reactivation. HSV reactivation in mice can be induced by various stimuli such as immunosuppression[20], hyperthermic stress[21],

or ultraviolet B irradiation[22], but these approaches induce minimal shedding, can be cumbersome, and may be applicable to only certain specific mouse strains or HSV isolates. Here we evaluated the reactivation of HSV by intraperitoneal injection of JQ1, a bromodomain inhibitor that has been proposed as a latency-reversing agent for human immunodeficiency virus (HIV)[23]. Previously, JQ1 was reported to reactivate HSV in cell culture models of latency[14] and induce shedding from the eyes of latently infected mice, although the amount and timing of shedding was not fully defined[14]. Here we demonstrate that JQ1 reproducibly induces detectable viral shedding at the periphery from a substantial subset of latently infected mice, at quantitative levels ($10^2$ to $10^6$ copies/mL) that are similar to those observed in

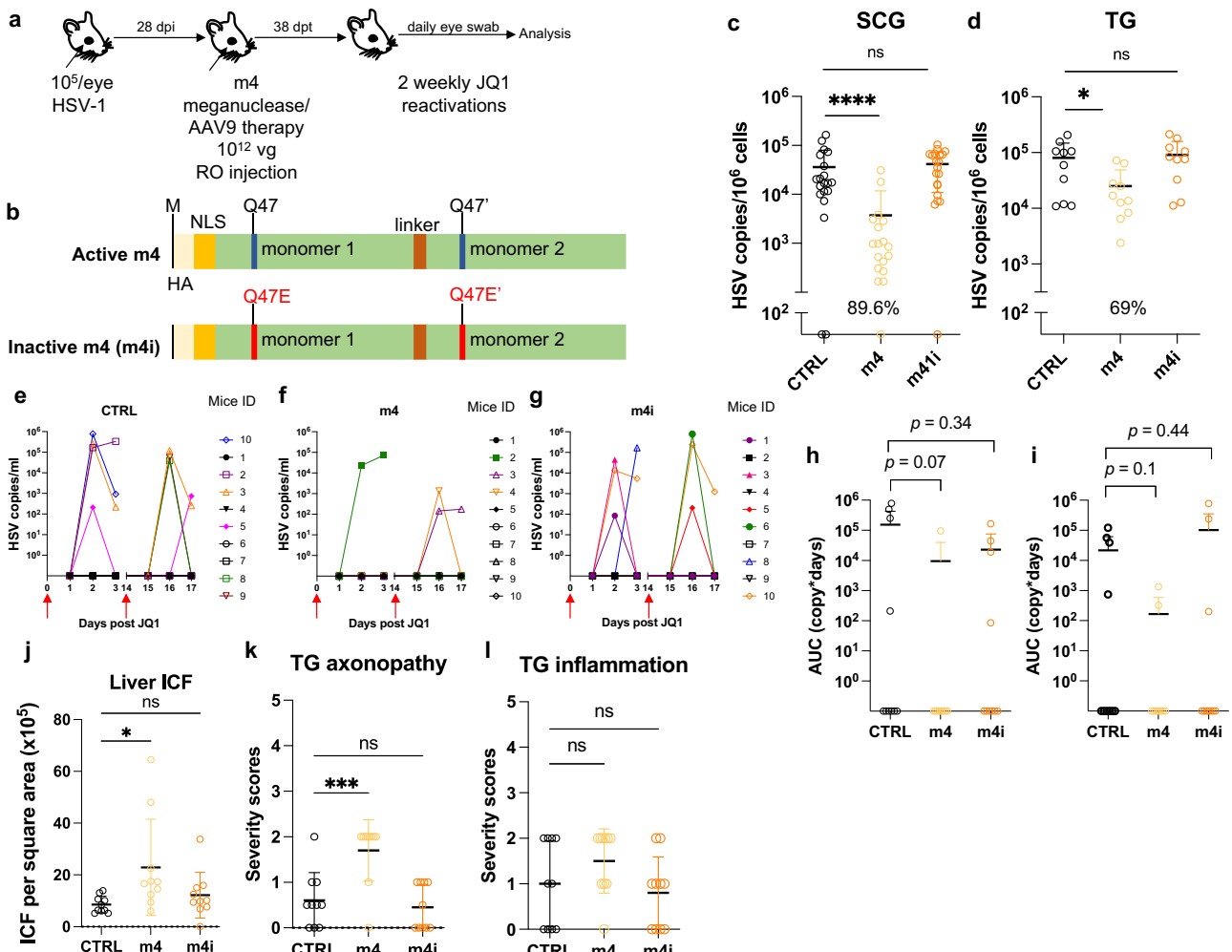

**Fig. 8 | Simplified meganuclease therapy decreases peripheral shedding in infected mice. a** Experimental timeline of ocular HSV-1 infection, meganuclease treatment and viral reactivations with JQ1. **b** Schematic of active m4 and inactive m4i meganuclease. **c-d**, HSV loads in both SCGs (**c**; p < 0.0001 for m4) and both TGs (**d**; p = 0.003 for m4) from control infected mice and infected mice treated the active m4 or inactive m4i (n = 10 per group). Percent decrease of HSV loads in treated mice compared to control mice and statistical analysis (unpaired one-tailed Mann-Whitney test with *p < 0.05; ****p < 0.0001; ns: not significant) are indicated. **e–g** Virus titers in eye swabs collected at day 1 to 3 after each JQ1 reactivation from control infected mice (**e**) and infected mice treated with active m4 (**f**) or inactive m4i (**g**). **h, i** Area under the curve (AUC) analysis of virus shedding after first (**h**), and

second (**i**) JQ1 reactivation from control infected mice and infected mice treated with active m4 or inactive m4i (n = 10 per group). p values (unpaired one-tailed Mann–Whitney test) are indicated. **j** Inflammatory cell foci (ICF) in liver sections from either HSV-infected control mice, mice treated with active m4 or inactive m4i (n = 10 per group); p = 0.0234 for m4. **k, l** Severity scores of axonopathy (**k**; p = 0.0007 for m4) and inflammation (**l**) in TG from HSV-infected control mice, mice treated with active m4 or inactive m4i (n = 10 per group) with statistical analysis (Ordinary one-way Anova, multiple comparisons with ns: not significant; *p < 0.05; ***p < 0.001). Each graph shows individual and mean values with standard deviation. AAV viral loads are shown in Supplemental Fig. 9m–o. Source data are provided as a Source Data file.

human observational studies[24]. Additional studies in C57BL/6 mice latently infected with HSV-1 showed that virus shedding could be also induced with JQ1 administration (Supplemental Table 4), suggesting that JQ1-induced virus shedding is not limited to the Swiss Webster strain. Thus, the JQ1 reactivation model should prove useful for future studies regarding the mechanisms and determinants of HSV reactivation and peripheral shedding, and vaccines or therapeutics aiming to reduce such shedding.

Among people infected with HSV, a major concern and driver of the desire for cure is the risk of transmission of the virus to others[9]. While our previous work demonstrated an up to 90% reduction of latent HSV within ganglia after gene editing[8], it remained unclear what effect such reduction would have on viral shedding at the periphery. Using the JQ1 reactivation model, we found that reduction of ganglionic load via gene editing has a profound effect on viral shedding, both in terms of the fraction of samples with detectable virus, and also in the amount of virus shed. In humans, the relationship between HSV

shedding quantity and the likelihood of viral transmission remains incompletely understood, but previous mathematical modeling suggests that reduction of shedding to levels below 10[4] viral copies (as observed in most of our treated animals that exhibited residual shedding) would be expected to greatly reduce, if not fully eliminate, the risk of viral transmission[25].

One challenge with the JQ1 reactivation model in mice is the stochastic nature of the induced viral reactivation and shedding. Within a given cohort, only a subset of JQ1-treated mice shed detectable HSV, and shedding in one episode was not predictive of subsequent shedding after repeated JQ1 reactivations. Similarly, individual mice often shed unilaterally from a single eye; again, this was not predictive of the laterality of subsequent shedding episodes. These findings are similar to observations in humans, in whom shedding is episodic and stochastic, and can occur at distinct anatomical locations during different shedding episodes[26]. However, they constituted a challenge in adequately powering experiments designed to detect a reduction in viral

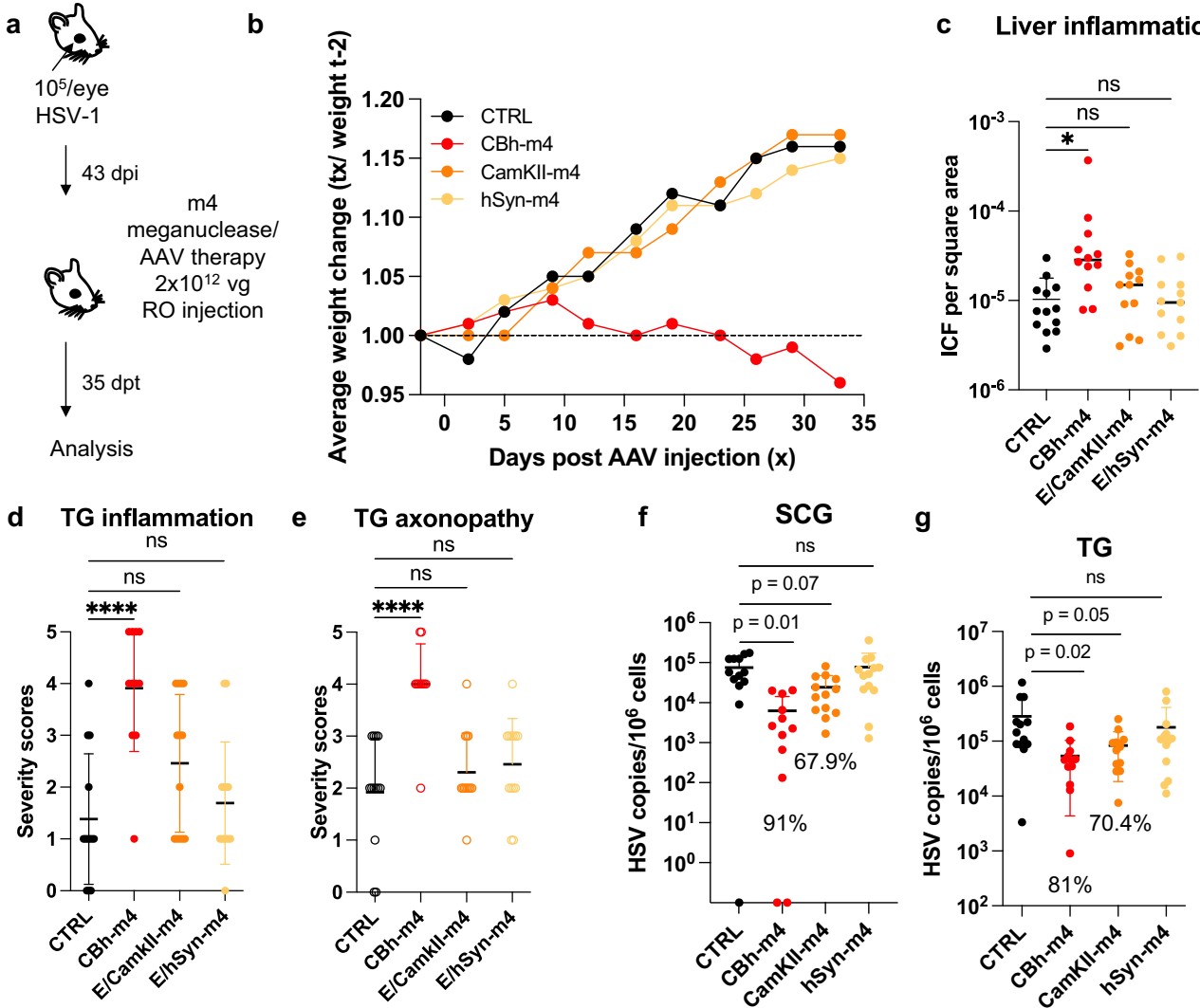

**Fig. 9 | a. Tissue restriction of meganuclease expression improves tolerability.**
**a** Experimental timeline of ocular infection and meganuclease therapy. **b** Average weight change of infected control mice (*n* = 13) or HSV-infected mice treated with m4 expressed from either the ubiquitous CBh promoter, or the neuronal promoters E/CamKII or E/hSyn (*n* = 12 per group). **c** Inflammatory cell foci (ICF) in liver sections from either HSV-infected control mice (*n* = 13), or mice treated with m4 expressed from either the CBh, E/CamKII or E/hSyn promoter (*n* = 12 per group); *p* = 0.00455 for CBh-m4. **d**, **e** Severity scores of inflammation (**d**; *p* = <0.0001 for CBh-m4) and axonopathy (**e**; *p* = <0.0001 for CBh-m4) in TG from infected control mice (*n* = 10) and infected mice treated with m4 expressed from either the CBh, E/

CamKII or E/hSyn promoter (*n* = 12 per group) with statistical analysis (Ordinary one-way Anova, multiple comparisons with ns: not significant; \**p* < 0.05; \*\*\*\**p* < 0.0001). **f**, **g** HSV loads in SCGs (**f**) and TGs (**g**) from infected control (*n* = 10) and infected mice treated with m4 expressed from either the CBh, E/CamKII or E/hSyn promoter (*n* = 12 per group). Percent decrease of HSV loads in treated mice compared to control mice and statistical analysis (Ordinary one-way Anova, multiple comparisons with *p* values; ns: not significant). Each graph shows individual and mean values with standard deviation. The AAV viral loads are shown in Supplemental Figure 10a–c. Source data are provided as a Source Data file.

shedding. While each of our individual experiments demonstrated a similar trend toward reduction of HSV shedding after AAV/meganuclease therapy, many individual experiments did not achieve statistical significance. This was particularly apparent after genital HSV infection, where the lower latent viral load in DRG appeared to result in lower reactivation rates after JQ1 administration. Optimization of the genital infection model to increase latent DRG loads may facilitate further evaluation of the effect of gene editing on viral shedding, although in our experience efforts to increase latent viral loads must be carefully balanced against increased animal mortality during the acute phase of infection. In any event, meta-analysis of the combined data from all our experiments confirmed a highly significant reduction in HSV shedding from AAV/meganuclease-treated animals compared with controls. This reduction in shedding proved to be both dose- and duration-of-therapy-dependent. The latter observation is particularly

encouraging in regard to the potential clinical translation of the work. Here, mice were evaluated for ganglionic load and shedding approximately one month after AAV/meganuclease administration, but our data suggest that HSV gene editing efficacy likely continues past this point, which may lead to more complete reduction or elimination of viral shedding at later time points. Experiments are currently underway to evaluate HSV gene editing in mice over periods of a year or more, and to determine whether overall efficacy ultimately plateaus.

A second important finding in our studies is that AAV-delivered meganucleases can readily enter neurons and edit HSV within DRG, the site of HSV latency in genital disease. We have consistently observed superior HSV viral load reduction in SCG (which are autonomic ganglia; typically ~90% reduction) compared to TG (sensory ganglia; typically ~50-60% reduction). This raised the possibility that DRG (also sensory ganglia) might also show similar modest rates of genome reduction.

However, gene editing of latent HSV genomes proved to be highly efficient within DRG (97% reduction), suggesting that the differing efficiencies are not intrinsic to the type of ganglion (sensory *vs.* autonomic). We currently favor the hypothesis that the relative efficiencies of AAV transduction in the various ganglia are driven mainly by the relative permeability of the blood/ganglionic barrier for each ganglionic type, and have a series of experiments underway to address this issue.

One limitation of our work is that the HSV-1 target sequences of the meganucleases used in this study are not well conserved in HSV-2. We therefore performed the vaginal infection experiments using HSV-1, to make the point that there appears to be no barrier to treating genital *vs.* orofacial disease; that is, the latently infected neurons in both sites are readily accessible to AAV vectors as well as meganuclease-mediated editing of latent HSV genomes. We would point out that successful treatment of genital HSV-1 infection is not a trivial result – over half of new cases of genital herpes in the US are now due to HSV-1[27]. Nevertheless, we fully recognize the importance of HSV-2 as a target, and current work in our laboratory focuses on development of anti-HSV-2 meganucleases and their testing in mice and other models of infection. We are especially interested in targeting duplicated regions of HSV-2, which our results here suggest will allow effective single-nuclease therapy, and we would suggest that duplicated or repeated sites should be considered for gene editing efforts targeting other viruses.

AAV vectors generally have been considered safe, particularly in comparison with other gene therapy vectors[28]. However, at high doses AAV vectors can induce liver toxicity, manifesting initially as transaminase elevation. At AAV doses higher than those used in the experiments presented here, liver toxicity can be severe, and has led to liver failure in several animal models[29,30]. We were therefore encouraged that we observed strong anti-HSV activity at AAV doses that were well tolerated in our mouse model. More recently, histological evidence of neuronal injury after AAV administration has been described in mice, rats, piglets, dogs, and non-human primates[29,31–33], and in at least two human trial participants at autopsy[34]. The causative mechanism of such injury remains unclear; among the current leading hypotheses are saturation of neural protein-folding capacity[35] and TLR9-mediated recognition of vector or transgene RNA[36]. Despite histological evidence of neuronal injury, clinical signs in experimental animals have been rare, consisting mainly of mild gait or balance disturbance[29,37,38]. Such signs have only been reported in a single patient among several thousand human participants in trials of AAV-delivered gene therapies[39]. Consistent with these other studies, we observed subtle evidence of neuronal injury in experimental mice, manifesting as neuronal degeneration, necrosis, and axonopathy, but our mice have not shown any associated behavioral alterations. Our results obtained using a neuronal-specific promoter to drive the meganuclease expression were somewhat surprising, in that they show an absence of the neuronal toxicity readily detected with use of an ubiquitous promoter. While not definitive in themselves, these results support consideration of an alternative hypothesis, that ganglionic neurotoxicity is mediated indirectly through AAV effects on non-neuronal cells, rather than on the neurons themselves. In any event, our data with regimen simplification, dose reduction, and tissue restriction of transgene expression is reassuring that mitigation avenues can be designed to eliminate both hepato- and neurotoxicity. Additional studies in alternative animal models of HSV infection, such as guinea pigs or even non-human primates, are therefore important, and if such studies confirm anti-HSV efficacy with an acceptable safety profile, human trials may be warranted. This is supported by the recent report of the use of a gene editing strategy to remove HSV from humans in an investigator-initiated, open-label, single-arm, non-randomized interventional trial in 3 patients with severe refractory herpetic stromal keratitis (HSK). In this study, no off-target cleavages or systemic adverse events were detected in the 18 months follow-up while preventing viral relapse[40].

## Methods

### Mice
In all studies, 5- to 8-week old female Swiss Webster mice were purchased from either Taconic or Charles River, and housed in accordance with the Fred Hutch Cancer Center and NIH guidelines on the care and use of animals in research. Experimental procedures performed and approved by the Institutional Animal Care and Use Committee (IACUC) of the Fred Hutch Cancer Center. Standard housing, diet, bedding, enrichment, and light/dark cycles were implemented under animal biosafety level 2 (ABSL2) containment.

A limited set of preliminary experiments was performed in C57BL/6 mice (Charles River), and we observed similar rates of viral load reduction and gene editing by T7 assay, suggesting that our results were not strain-specific.

### Ocular HSV infection
Mice were anesthetized by intraperitoneal injection of ketamine (100 mg/kg) and xylazine (12 mg/kg). Mice were infected in both eyes by dispensing $10^5$ PFU of HSV1 syn17+ contained in 4 ul following corneal scarification using a 28-gauge needle.

### Vaginal HSV infection
Mice were treated with 2 mg of Depo-Provera injected subcutaneously. Five to seven days later, they were anesthetized by intraperitoneal injection of ketamine (100 mg/kg) and xylazine (12 mg/kg) and intravaginally infected with either $5 \times 10^2$ or $10^3$ PFU of HSV1 syn17+ contained in 4 ul using a pipette after clearing the vaginal lumen with a Calginate swab.

### AAV inoculation
Mice anesthetized with isoflurane were administered the indicated AAV vector dose by either unilateral intradermal whisker pad (WP), or intravenous injection using unilateral retro-orbital (RO; ocular HSV-infected mice) or tail vein (TV; ocular and vaginal HSV-infected mice) injection. Tissues were collected at the indicated time.

### Study approval
All animal procedures were approved by the Institutional Animal Care and Use Committee of the Fred Hutchinson Cancer Center. This study was carried out in strict accordance with the recommendations in the Guide for the Care and Use of Laboratory Animals of the National Institutes of Health ("The Guide").

### HSV reactivation
JQ1 reactivation was performed by intraperitoneal (IP) injection of (+)-JQ-1 (JQ1, MedChemExpress) at a dose of 50 mg/kg at the indicated time. JQ1 was prepared from a stock solution (50 mg/ml in DMSO, Sigma) by 1:10 dilution in a vehicle solution 10% w/v 2-hydroxypropyl-β-cyclodextrin in PBS (Sigma). Alternatively, JQ1 reactivation was performed with 2 IP injections separated by 12 h, which resulted in the detection of virus shedding in the eyes of 6 out of 9 (66.6%) mice (Supplemental Fig. 5). A limited set of experiments was performed in C57BL/6 mice (Charles River), and we observed similar rates and kinetics of peripheral viral load shedding, suggesting that the results of JQ1 reactivation were not mouse strain specific. Hyperthermic stress (HS) reactivation was performed as previously described[15].

### Cells, herpes simplex viruses, and AAV stocks
HEK293[41] and Vero cell lines (ATCC # CCL-81) were propagated in Dubelcco's modified Eagle medium supplemented with 10% fetal bovine serum. HSV-1 strain *syn*17 + (kindly provided by Dr. Sawtell) was propagated and titered on Vero cells.

AAV production and titering. AAV vector plasmids pscAAV-CBh-m5, pscAAV-CBh-m8, pscAAV-CBh-m4, pscAAV-CBh -m4i and pscAAV-E/CamKII-m4 were used to generate the AAV stocks in this study

(Fig. S12). All AAV stocks were generated from transfected HEK293 cells and culture media produced by the Viral Vector Core of the Wellstone Muscular Dystrophy Specialized Research Center (Seattle). AAV stocks were generated by PEG-precipitation of virus from cell lysates and culture media, followed by iodixanol gradient separation[42,43] and concentration into PBS using an Amicon Ultra-15 column (EMD Millipore). AAV stocks were aliquoted and stored at −80ºC. All AAV vector stocks were quantified by qPCR using primers/probe against the AAV ITR, with linearized plasmid DNA as a standard, according to the method of Aurnhammer et al.[44]. AAV stocks were treated with DNase I and Proteinase K prior to quantification.

### Quantification of viral loads in tissues

Total genomic DNA was isolated from ganglionic tissues using the DNeasy Blood and tissues kit (Qiagen, Germantown, MD) per the manufacturer's protocol. Viral genomes were quantified by ddPCR in tissue DNA samples using an AAV ITR primer/probe set for AAV, and a gB primer/probe set for HSV as described previously[7]. Cell numbers in tissues were quantified by ddPCR using mouse-specific RPP30 primer/probe set: Forward 5′-GGCGTTCGCAGATTTGGA, Reverse 5′-TCCCAGGTGAGCAGCAGTCT, probe 5′-ACCTGAAGGCTCTGCGCG-GACTC. In some control ganglia, sporadic samples showed positivity for AAV genomes, although the levels were typically >2-3 logs lower than in ganglia from treated mice having received AAV. We attribute this to low-level contamination of occasional tissue samples. The ganglionic AAV loads for experiments presented in Figs. 1–3 and 5–8 are shown in Supplemental Fig. 9 and those for the experiment presented in Fig. 9 are shown in Supplemental fig. 10. The statistical analysis was performed using GraphPad Prism version 9.4.1. The test used for each data set is indicated in the figure legends.

### Quantification of HSV in eye swabs

Swab samples were collected into vials containing 1 ml of digestion buffer (KCL, Tris HCl pH8.0, EDTA, Igepal CA-630). DNA was extracted from 200 ml of digestion buffer using QiaAmp 96 DNA Blood Kits (Qiagen, Germantown, MD) and eluted into 100 ml AE buffer (Qiagen). Then, 10 ml of DNA was used to setup 30 ml real-time Taqman quantitative PCR reactions. The primers and probes were as described previously[45]. QuantiTect multiplex PCR mix (Qiagen) was used for PCR assays. The PCR cycling conditions were as follows: 1 cycle at 50 °C for 2 min, 1 cycle at 95 °C for 15 min, and 45 cycles of 94 °C for 1 min and 60 °C for 1 min. Exo internal control was spiked into each PCR reaction to monitor inhibition. A negative result was accepted only if the internal control was positive with a cycle threshold (CT) within 3 cycles of the Exo CT of no template controls.

### Western blot detection

Tissue lysates were obtained from 1 TG per mouse collected in 200 µl RIPA buffer (PIERCE, Thermo Scientific) with protease inhibitor cocktail (Roche) and disrupted by sonication on ice. Thirteen microliters of tissue lysates were loaded onto 4-12% NuPAGE gel, transfer onto nitrocellulose membrane and probe for m4 expression using rabbit anti-HA antibody (1:1000 mAb clone C29F, Cell signaling) and β-actin (1: 1000 mAb clone13E5, cell signaling) for protein loading. Membrane hybridization and detection were performed using PIERCE Fast Western blotting kit Super signal, West pico Rabbit (Thermo Scientific) per manufacturer protocol and imaged using ChemiDoc Imaging system (BIO-RAD).

### Inflammatory cell foci quantification

Liver tissues were paraffin-embedded, sectioned and H&E stained by the Experimental histopathology shared resources of the Fred Hutchinson Cancer Center. ICF were counted by a blinded observer and expressed as the number of ICF per surface area which was determined using Fiji[46].

### Grading of neuronal changes within trigeminal ganglia

Trigeminal ganglia were paraffin-embedded, sectioned, and H&E stained by the Experimental histopathology shared resources of the Fred Hutchinson Cancer Center. Microscopic changes were graded as to severity by a veterinary pathologist using a standard grading system whereby 0 = no significant change, 1 = minimal, 2 = mild, 3 = moderate, and 4 = severe.

### Statistical analysis

Statistical analyses for each individual experiment were performed using GraphPad Prism version 9.4.1 and R. Tests were two-sided and p-values smaller than 0.05 considered significant. The specific test used for each analysis is indicated in the corresponding figure legend.

Meta-analyses were performed on combined data from all experiments (Figs. 1–6). To assess whether AAV/MN-treated mice shed virus less often than control animals and whether the frequency of viral shedding decreased with the duration of meganuclease therapy, we used generalized linear mixed models (GLMM) describing the probability of viral shedding (with viral shedding defined as an AUC > 0) as a function of therapy duration and dose treated as continuous variables, while adjusting for experiment. Animal-specific random intercepts were included in the model to capture intra-mice dependencies between observations. We also considered including an interaction term between dose and duration to evaluate whether change in the probability of viral shedding over time was affected by dose (in particular, whether it decreased faster with dose). Association between viral shedding and covariates (dose, therapy duration) are reported as odds ratios (OR). The significance of the association between the probability of viral shedding and dose, therapy duration, and their interactions was evaluated using two-sided Wald tests.

To study the relationship between the quantity of virus shedding and therapy dose and duration, we used linear mixed models (LMM) describing the log10-transformed AUC as a function of therapy dose and duration, treated as continuous variables, while adjusting for experiment. The model included an interaction between dose and therapy duration to assess whether change in AUC over time was impacted by dose (e.g., whether the log10-transformed AUC decreased faster with dose). Animal-specific random intercepts were also included to capture intra-mice dependencies between observations. The standard errors of regression coefficients were estimated using a robust, sandwich-type estimator. Association between viral shedding and covariates (dose, therapy duration) are reported as odds ratios (OR). The significance of the association between the log10-transformed AUC and dose, therapy duration, and their interactions was evaluated using Wald tests. Both sets of analyses also evaluated models that included dose squared and square root of dose to perform sensitivity analysis and assess whether the relationship between the probability of viral shedding (or the log10-transformed AUC) and dose were nonlinear. Likewise, models were considered that included squared therapy duration or square root of therapy duration to perform sensitivity analysis and assess whether relationships between the probability of shedding (or the log10-transformed AUC) and therapy duration were nonlinear.

### Reporting summary

Further information on research design is available in the Nature Portfolio Reporting Summary linked to this article.

## Data availability

All data generated are provided in the main text, the supplementary materials or the Source Data file. Meganuclease sequences are Cellectis proprietary information and material. Source data is provided with this paper. Source data are provided with this paper.

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

## Acknowledgements

We are grateful to Cellectis (Paris, France) for the original development of meganucleases m4, m5, and m8, as well as Philippe Duchateau and Roman Galetto (Cellectis) for helpful discussions. This work was supported by National Institutes of Health grant R01AI132599 (KRJ), Caladan Foundation (KRJ), National Institutes of Health grant 5P50AR065139 (Viral Vector Core of the Wellstone Muscular Dystrophy Specialized Research Center, Seattle), National Institutes of Health / National Cancer Institutes, Cancer Center Support Grant P30 CA015704 (Fred Hutchinson Cancer Center Shared Resources Division), donations from the Kenneth Hill Foundation, Krieger Family Trust, Tiny Foundation, and over 2000 individual donors.

## Author contributions

Conceptualization: K.R.J., M.A., D.S., D.E.S. and O.H. Investigation: M.A., A.K.H., D.E.S., L.M.K., M.A.L., L.S., T.K.S. and O.H. Funding acquisition: K.J.R., M.A. and D.S. Supervision: K.R.J., M.A., D.S. and M.L.H. Writing – original draft: M.A., K.R.J. and O.H. Writing – review & editing: M.A., K.R.J., D.S. and O.H.

## Competing interests

K.R.J. is a founder and holds equity in Caladan Therapeutics, is a paid advisor and holds equity in Excision Biosciences, participates in sponsored research agreements with Excision Biosciences and Emendo Biotherapeutics, and is co-inventor of International Patent Application No. PCT/US2022/013757 and U.S. Provisional Application No. 63/503,541 held by Fred Hutch for the treatment of HSV-1 and HSV-2 using meganucleases. There are no restrictions on publication of data. M.A. and D.S. have sponsored research agreements with Excision Biosciences. The remaining authors declare no conflict of interest.
