## [Peer Review File · Nature Communications]

Gene editing for latent herpes simplex virus infection reduces viral load and shedding in vivoReviewer #1 (Remarks to the Author):

The article by Aubert et al is a pretty good paper. It details the extension of studies of considerable interest that were first reported in a previous nature communications publication (reference 8) in which the aim is to reduce the shedding and genome load of HSV in the murine models of ocular and vaginal infection by targeting the latent viral DNA using genome editing methods. They appear to have greatly improved the approach and have achieved considerable reductions in ganglionic HSV latent loads using combinations of optimized different serotype AAV, expressing multiple meganucleases following retroorbital or intradermal nasla inoculation. Globally, the data is quite convincing and would be important for the field in moving towards a possible curative step of removal of latent HSV to reduce disease from reactivations.

They studies include optimization of serotypes, the use of multiple meganucleases to HSV and accurate quantification of ganglionic HSV genome loads after treatment. One comment is that it would be highly useful to know if these strategies are mouse strain specific, or if they work in other mice strains, such as the widely used B6 or Balb-c strains used by others?

An additional and thoughtful study examines hepatic pathology, as some AAV therapies have indicated hepatitis toxicity. This seems like a sidetrack, given the stage of these studies, but they are supplemental and not part of the main message, and as just pointed out, are important to know.

Another important advance in the article is the improved reactivation frequency they obtain with the Bromodomain inhibitor JQI, which is demonstrated to be more efficient than heat stress at inducing reactivation and, more importantly, virus levels of shedding at the eye (which is acknowledged to be difficult and quite variable). A note of question- how do their reactivation frequencies with heat stress compare with those of the Sawtell/thompson group?. Are the low levels seen here for heat stress comparable to those seen by the sawtell group? And does JQI work in other mice strains? This would be very important information for the HSV murine field to know.

A couple of other concerns do arise. The vaginal model studies are not as fully convincing, particularly in regards to reactivation stimulated by JQI- even in the controls, only one or two mice showed shedding, and one mouse in the treated group shows shedding. This data is considered stretching the data a bit too far and does not justify inclusion or analyses with the eye work, and perhaps should be dropped from the study, given that only two mice reactivated to give shedding in the control group for a few days, while one shed in the treated group for a day (Figure 5) The fact is that nearly all mice in each group showed no shedding. Conclusions are too overreached based on these minimal less than convincing datasets (although the reduction in ganglionic genome load is convincing). The reactivation data detracts from the convincing nature of the ocular studies that are predominantly focused on in this work.

Minor points

Line 23 the source of recurrent disease. Also I would call the strategy potentially curative; there is still a lot of HSV genomes in the ganglia after AAV meganuclease treatment. Curative is a word I would use when there is none in the ganglia, 97% is not very big when you are starting with millions of copies of viral DNA

Line 29. Limit this statement to ocular infected mice as the vaginal model shedding data is poor

Line 32 again extending the data to the genital infections would need better data than that in the paper as it stands. Genome load is reduced, yes. It is not clear from the data if this is effective at reducing vaginal shedding.

Line 53 meaning what? genome loads? Reactivable virus? Titers upon reactivation?

Line 44 I am surprised the list does not include ocular lesions, which can be devastating, and indeed, the majority of the paper concerns a murine ocular infection model

Line 64 The vaginal model work- are the HSV-2 sites targeted by the meganucleases and are they identical to that in HSV-1? Thus is the strategy effective for HSV-2? This is likely more important than HSV-1, and HSV-2 shedding is more frequent and more subtle and a more important target

A general comment. Does this work establish any support for non trigeminal ganglia causing reactivated disease? Since AAV is so much more effective in the SCG than TG.....

Line 95 should mention in the text that infection was with Depo Prevara treatment. Also figure 1a shows schematic of ocular infection; however part b concerns vaginal infection, so the diagram is only

partial. It is thus unclear where AAV were inoculated when looking for vaginal model efficacy. This bit is thus a little confusing to the first time thru reader, as a person might think the eye is the route for the treatment in the vaginal model. Please make the studies more clearly explained as to what was actually done

Lin 107 define BET, it is not a common phrase

Line 129 could this be interpreted to support that a large portion of the ganglionic load represents unreactivable genomes (under these conditions?). This has long been speculated and may reflect behaviors of different neuronal subtypes to reactivation stimuli.

Line 277 this is quite similar to VZV reactivations in that a prior event is not a predictor of downstream events. Could this be interpreted to mean that perhaps, the reactivated neuron does not survive the reactivation event?

Line 337 not quite clear how and where AAV were introduced for vaginal infection and blocking of shedding studies

Reviewer #2 (Remarks to the Author):

The authors developed an approach against HSV infection, based on gene editing using HSV-specific meganucleases delivered by adeno-associated virus (AAV) vectors, which is composed of two anti-HSV-1 meganucleases delivered by a combination of AAV9, AAV-Dj/8, and AAV-Rh10 and showed elimination of latent HSV DNA in both ocular and vaginal mouse models of HSV infection. My one of my main concerns is it's novelty.

I compared this manuscript with the Nature Communication paper published by the same group in 2020.

The claims of novelty:

1. High efficiency using triple AAV (9, Dj/9, Rh10) infection. But this triple AAV (1+8+Rh10) strategy was used in the previous paper and achieved similar efficiency to clear HSV genomes in SCG and TG peripheral neurons. In addition, the authors used a much higher total AAV dose (3×10^{12} vg/mouse) in this study, which induced toxicity in mice. If the lower dose (1.8×10^{12} vg/mouse) was used (Figure 4a-c), the HSV genome clearing efficiency was actually lower than that achieved in their previous study.

2. Established HSV reactivation/shedding model with QJ-1 drug induction. But this model is highly variable. With such high variations, authors didn't do a sample size calculation to determine the animal number for each expt so for some key expts the result were not statistically significant (Figure 2f, 2l, 3j-l, 4g).

Minor issues:

Figure 3g missing

Figure 2j difficult to see the individual mouse curve without coloring.

Reviewer #3 (Remarks to the Author):

Keith Jerome's lab presents an interesting follow-up to a gene editing approach to "curing" HSV-1 latent virus from the trigeminal and superior sacral ganglia following eye inoculation and the dorsal root ganglia following intravaginal inoculation. Overall the authors eliminated 97% of "latent" HSV DNA in ganglia. This reduction translated into partial reductions in reactivated virus following treatment with the BET bromodomain inhibitor JQ1 to induce HSV-1 reactivation from latency. The interference with virus reactivation differed substantially in treatment of different ganglia; TG being the least controlled. This strategy might be translated to patients but issues remain regarding methods of delivery, AAV transduction efficiency and vector toxicity. It is not clear why AAV is being developed and not HSV that will naturally enter the same neuronal cell types without toxicity and one

HSV vector can easily be used to express all three nuclease transgenes. Perhaps the authors would like to discuss.

Comments:

1. It would be of significant interest to know the efficiency of uptake of the AAV vectors into sensory neurons and to what extent AAV genomes overlap with latent HSV. In their previous publication, scRNA analysis was used to address this issue. Why wasn't this type of study repeated since one goal here was to enhance transduction efficiency. Since TG treated animals show inefficient blocking of HSV reactivation, is this due to a lack of co-localization of AAV and HSV in neurons particularly since multiple AAV vectors are in use. Three different meganuclease expressing vectors are presumably needed to create multiple single stranded breaks in the HSV genome possibly to prevent possible HSV DNA repair. The nuclease cDNA is likely about 3kb that necessitates three different vectors, an expensive and complex strategy for manufacture? Is the assumption that the different AAV vectors infect the same or different neuronal cell populations? I suspect that the distribution of AAV differs substantially in different test applications which accounts for why some animals are unaffected by the AAV treatment. Delivery remains an important high bar for effective therapy.
2. Two anti-HSV-1 meganucleases were delivered using a combination of AAV vectors (AAV9, AAV-Dj/8 and AAV-Rh10) to attack resident HSV DNA. What are the specific targets? Earlier work targeted VP5, polymerase and ICP0. There are no data on meganuclease expression in vivo or HSV DNA mutations.
3. Because initial virus shedding from the TG was not affected by AAV expression of nucleases but reduced substantially in later drug inductions, how will this be useful to treatment of human disease. Generally human reactivations are sporadic.
4. The reduction of HSV genomes in the TG would seem to be most important given the clinical symptoms of herpes keratitis. Fig 1 shows a reduction of 61% and 42% reduction in latent virus but no statistically significant reduction in TG reactivation (Fig. 2). What accounts for this difference. Is there evidence of inflammation of the cornea?
5. Among controls and after JQ1 treatment, how many neurons show active HSV transcription of a standard lytic gene such as glycoprotein D compared with attacked viral gene.
6. Do the authors know how many neurons show virus reactivation. In natural infections, this number could be very low which speaks to the importance of efficient co-localization of AAV and HSV in neurons. Drug treatment may activate many neurons to release HSV but in human infections this could be limited and hit or miss as to whether AAV has inactivated HSV in reactivating neurons.
7. Since the reduction in HSV virus load differed substantially between SCG and TG, does this reflect AAV transduction efficiency of HSV containing neurons? What are the kinetics of AAV transgene expression. It might be expected to come on slowly and peak after weeks. How might this affect efficiency of KD of HSV genomes? The data suggest that the treatment improves over time due AAV based gene expression however in applications in which the timing of AAV administration will be difficult to assess how might this be dealt with? The authors need to address this in the discussion.
8. The HSV inoculates are at one dose. There needs to be a dosing experiment to assess HSV DNA destruction efficiency. How does the distribution and amount of HSV DNA correlate with the number of genome copies per neuron and the impact on AAV-mediated genome reduction. The number of latent genome copies per cell is known to vary considerably from a few to thousands of copies. Will high copy DNA neurons be resistant to treatment?
9. Using only HSV-1 as a model system in vaginal infections does not adequately reflect the fact that the majority of vaginal infections are HSV-2 which may be more difficult to knock down. Do the selected meganucleases destroy HSV-2 DNA? Guinea pigs would be a more adequate model system since natural recurrence occurs in these animals.
10. Following vaginal infections, which DRG are harboring HSV?.
11. I assume that AAV will persist in various tissues including sensory neurons. Over time one might expect the meganuclease protein to induce an immune response. Have authors looked for this?
12. About a third of individuals have anti-AAV neutralizing antibodies. Will this anti-vector immune activity affect the efficiency of transduction of sensory neurons?
13. Delivery of AAV to sensory could be highly problematic and potentially cytotoxic as observed in this study. How do the authors envision vector delivery in a human application? High dose AAV

delivery to ganglia in primates has recently been shown to be highly cytotoxic so safety will be an issue (Hum Gene Ther. 2018 Mar;29(3):285-298). Toxicity was independent of the transgene and virus capsid. The authors indicated that they did not see detectable signs of neuropathy following vector treatment however there was evidence of neurodegeneration. I would imagine that this could lead to neuropathic pain. This potential safety problem needs to be evaluated.

14. How does AAV therapy compare with ganciclovir treatment, the standard of care regarding reactivated virus load?

RESPONSE TO REVIEWER COMMENTS

We thank the reviewers for their thoughtful review and helpful suggestions to improve the manuscript. We have performed a substantial amount of new experimentation to address their comments, as detailed below, and incorporated their suggestions into the revised manuscript.

Our point-by-point responses to the reviewers' comments are as follows:

Reviewer #1 (Remarks to the Author):

The article by Aubert et al is a pretty good paper. It details the extension of studies of considerable interest that were first reported in a previous nature communications publication (reference 8) in which the aim is to reduce the shedding and genome load of HSV in the murine models of ocular and vaginal infection by targeting the latent viral DNA using genome editing methods. They appear to have greatly improved the approach and have achieved considerable reductions in ganglionic HSV latent loads using combinations of optimized different serotype AAV, expressing multiple meganucleases following retroorbital or intradermal nasla inoculation. Globally, the data is quite convincing and would be important for the field in moving towards a possible curative step of removal of latent HSV to reduce disease from reactivations.

They studies include optimization of serotypes, the use of multiple meganucleases to HSV and accurate quantification of ganglionic HSV genome loads after treatment.

One comment is that it would be highly useful to know if these strategies are mouse strain specific, or if they work in other mice strains, such as the widely used B6 or Balb-c strains used by others?

Response:

We agree that this is an important point, and initially chose to use the outbred Swiss-Webster mouse for our experiments to minimize the possibility of strain-specific effects. In response to the reviewer's question, we performed a limited set of preliminary studies in C57BL/6 mice, and observe similar rates of viral load reduction and gene editing by T7 assay as in Swiss-Webster mice, suggesting that the results are not strain-specific. We added a mention of this in the materials and methods section:

Page 19, lines 427-429: "A limited set of preliminary experiments was performed in C57BL/6 mice (Charles River), and we observed similar rates of viral load reduction and gene editing by T7 assay, suggesting that our results were not strain-specific."

An additional and thoughtful study examines hepatic pathology, as some AAV therapies have indicated hepatitis toxicity. This seems like a sidetrack, given the stage of these studies, but they are supplemental and not part of the main message, and as just pointed out, are important to know.

Response:

In general, the consensus feedback we have received regarding our work is that safety studies are indeed important, as the reviewer states. Reviewer 3 also raises important questions about safety. We believe that it is important to evaluate safety in parallel with our efficacy studies, and that this approach is the most efficient use of animals and other resources. In response, we have extended our safety studies and now demonstrate that undesirable effects on both the liver and ganglia can be essentially eliminated through regimen simplification (to a single AAV serotype and

single meganuclease), dose reduction, and tissue restriction of meganuclease expression. These data have been added to the manuscript in two new subsections of the results, “Simplification of the AAV-meganuclease regimen” and Tissue restriction of meganuclease expression improves tolerability”, found on page 11, line 245, through page 14, line 300. We believe that these results greatly improve the likelihood of clinical translation of our strategy, and that they will be of substantial interest to others in the gene therapy field.

Another important advance in the article is the improved reactivation frequency they obtain with the Bromodomain inhibitor JQ1, which is demonstrated to be more efficient than heat stress at inducing reactivation and, more importantly, virus levels of shedding at the eye (which is acknowledged to be difficult and quite variable). A note of question- how do their reactivation frequencies with heat stress compare with those of the Sawtell/thompson group?. Are the low levels seen here for heat stress comparable to those seen by the sawtell group?

Response:

It is difficult to compare the previously published results from Sawtell/Thompson to ours, as the Sawtell/Thompson papers defined reactivation based on the detection of infectious viruses within in the trigeminal ganglia (see Sawtell et al 1998; Sawtell 2004; Sawtell 2005), while we evaluated for virus shed at the periphery via PCR on eye swabs. Sawtell/Thompson reported the peak of infectious virus detected in the ganglia after heat stress was at 22-24h post reactivation, which seems consistent with our observation of peak virus shedding at the periphery after JQ1 administration at about 48h post reactivation. Sawtell/Thompson reported the frequency of reactivation as measured by the detection of infection viruses in TG after heat stress to be 60-70%, while we observed the frequency of peripheral virus shedding following JQ1 reactivation to be 30-55%, which again seems generally consistent. While a direct comparison of the two methods of viral reactivation is not the main point of our paper, we did directly compare the two methods (heat shock vs. JQ1) for their ability to induce peripheral viral shedding, and find that JQ1 induces both a higher frequency of shedding along with higher peak peripheral viral loads. To our knowledge there are no additional studies directly linking the detection of infectious viruses in ganglionic tissues with peripheral virus shedding, and it remains unknown whether the presence of infectious viruses in neurons always leads to detectable viruses at the periphery in vivo, although our results suggest that it may not. We have included these findings in the results section:

Page 6, lines 124-127: “A direct comparison suggested that JQ1 may be a more powerful reactivation stimulus for HSV than hyperthermic stress (HS), which in our hands led to detectable virus shedding in less than 20% of animals (2/12 HS vs 4/10 JQ1), with peak shedding viral loads two logs lower than after JQ1 treatment (Fig. S3d-f).”

And does JQ1 work in other mice strains? This would be very important information for the HSV murine field to know.

Response:

We appreciate the reviewer pointing out the value this information would have to the HSV murine field. We therefore evaluated the effect of JQ1 in HSV-infected C57BL/6 mice. We observe similar rates and kinetics of peripheral viral shedding as in Swiss-Webster mice, suggesting that the results are not strain-specific. We added a mention of this in the materials and methods section:

Page 19, lines 427-429: "A limited set of experiments was performed in C57BL/6 mice (Charles River), and we observed similar rates and kinetics of peripheral viral load shedding, suggesting that the results of JQ1 reactivation were not mouse strain specific."

A couple of other concerns do arise. The vaginal model studies are not as fully convincing, particularly in regards to reactivation stimulated by JQ1- even in the controls, only one or two mice showed shedding, and one mouse in the treated group shows shedding. This data is considered stretching the data a bit too far and does not justify inclusion or analyses with the eye work, and perhaps should be dropped from the study, given that only two mice reactivated to give shedding in the control group for a few days, while one shed in the treated group for a day (Figure 5) The fact is that nearly all mice in each group showed no shedding. Conclusions are too overreached based on these minimal less than convincing datasets (although the reduction in ganglionic genome load is convincing). The reactivation data detracts from the convincing nature of the ocular studies that are predominantly focused on in this work.

Response:

We appreciate this feedback, and have taken several steps to temper our interpretation of these studies, particularly in regard to the shedding data. We agree that the observed shedding rates after JQ1 administration were lower in the genitally infected mice compared with ocularly infected mice, and believe this relates to the substantially lower latent HSV loads (~2 logs) in DRG relative to TG or SCG. Based on this comment as well as the minor comments below, we have modified lines 29 and 34 in the abstract to remove any reference to genital disease.

We have revised the section regarding genital shedding more accurately reflect the limitations of our data:

Page 10, lines 204-214: "We then sought to evaluate whether JQ1 could induce HSV shedding in the genital infection model, as we previously observed in the ocular infection model. Over 3 sequential JQ1 reactivations, only 2 of 8 control animals (and 1 of 8 AAV/meganuclease treated animals) shed detectable virus, a rate lower than the 40-50% reactivation we typically observe after ocular infection. The apparently lower rate of reactivation seen in the vaginal model compared to the ocular model may be due to lower levels of ganglionic viral loads in the DRG (10^2 - 10^3 vg/ 10^6 cells in DRG, Fig. 4i vs 10^4 - 10^5 vg/ 10^6 cells in SCG or TG, Fig. 4b-c). While this lower reactivation rate prohibited meaningful statistical analysis, the observation that 2 out of the 8 control mice shed virus over 2 to 3 sequential days, while only 1 of the 8 AAV-treated mice shed virus, on a single day and at a substantially lower level, is qualitatively in agreement with our observations after ocular infection (Fig. 4j-l)."

Finally, we modified the discussion to add mention of the reduced shedding observed after genital infection, the most likely explanation, and possible improvements to the experimental system:

Page 16-17, lines 359-364: "This [the stochastic nature of induced shedding] was particularly apparent after genital HSV infection, where the lower latent viral load in DRG appeared to result in lower reactivation rates after JQ1 administration. Optimization of the genital infection model to increase latent DRG loads may facilitate further evaluation of the effect of gene editing on viral shedding, although in our experience efforts to increase latent viral loads must be carefully balanced against increased animal mortality during the acute phase of infection."

Minor points

Line 23 the source of recurrent disease. Also, I would call the strategy potentially curative; there is still a lot of HSV genomes in the ganglia after AAV meganuclease treatment. Curative is a word I would use when there is none in the ganglia, 97% is not very big when you are starting with millions of copies of viral DNA

Response:
Change made.

Line 29. Limit this statement to ocular infected mice as the vaginal model shedding data is poor

Response:
See response to final major comment above.

Line 32 again extending the data to the genital infections would need better data than that in the paper as it stands. Genome load is reduced, yes. It is not clear from the data if this is effective at reducing vaginal shedding.

Response:
See response to final major comment above.

Line 53 meaning what? genome loads? Reactivable virus? Titers upon reactivation?

Response:
We have clarified the statement as follows:

Page 3, lines 53-55: "In a recent study, AAV-delivered meganucleases eliminated over 90% of HSV-1 genomes from the superior cervical ganglia of latently infected mice."

Line 44 I am surprised the list does not include ocular lesions, which can be devastating, and indeed, the majority of the paper concerns a murine ocular infection model

Response:
This is an excellent point, and we added mention of ocular lesions as follows:

Page 3, lines 45-46: "HSV infections can cause recurrent orofacial, corneal, anogenital, or other lesions, and infections of newborns can lead to disseminated disease and devastating neurological sequelae."

Line 64 The vaginal model work- are the HSV-2 sites targeted by the meganucleases and are they identical to that in HSV-1? Thus is the strategy effective for HSV-2? This is likely more important than HSV-1, and HSV-2 shedding is more frequent and more subtle and a more important target

Response:

Unfortunately, the HSV-1 target sequences of the meganucleases used in this study are not well conserved in HSV-2. We performed the vaginal infection experiments to make the point that there appears to be no barrier to treating genital vs. orofacial disease; that is, the latently infected neurons in both sites are readily accessible to AAV vectors as well as meganuclease-mediated editing of latent HSV genomes. We would point out, however, that successful treatment of genital HSV-1 infection is not a trivial result – over half of new cases of genital herpes in the US are now due to HSV-1. Nevertheless, we agree with the reviewer that genital HSV-2 shedding is more frequent, and that HSV-2 may be a more subtle and important target. We have rationally engineered a series of anti-HSV-2 meganucleases that are currently being tested in vitro and in vivo. However, that work is not yet mature enough for publication, and will be the subject of a future report. We have added a paragraph addressing these points to the discussion section:

Page 18, lines 387-395: “One limitation of our work is that the HSV-1 target sequences of the meganucleases used in this study are not well conserved in HSV-2. We therefore performed the vaginal infection experiments using HSV-1, to make the point that there appears to be no barrier to treating genital vs. orofacial disease; that is, the latently infected neurons in both sites are readily accessible to AAV vectors as well as meganuclease-mediated editing of latent HSV genomes. We would point out that successful treatment of genital HSV-1 infection is not a trivial result – over half of new cases of genital herpes in the US are now due to HSV-1²⁷. Nevertheless, we fully recognize the importance of HSV-2 as a target, and current work in our laboratory focuses on development of anti-HSV-2 meganucleases and their testing in mice and other models of infection.”

A general comment. Does this work establish any support for non trigeminal ganglia causing reactivated disease? Since AAV is so much more effective in the SCG than TG

Response:

We very much appreciate this fascinating question, which we have contemplated and debated for some time. We generally subscribe to the view that latent virus in autonomic ganglia, such as the SCG, may contribute to HSV reactivation, while recognizing that there is little direct literature shedding light on the issue. Our results are generally consistent with this notion, since viral load reduction with meganuclease therapy is consistently better in SCG than in TG, and yet at the same time there is a clear effect on viral shedding. Nevertheless, our experiments were not designed to address this issue, and we realize that alternative interpretations of our data exist. Thus, we chose not to discuss this point in the manuscript.

Line 95 should mention in the text that infection was with Depo Prevara treatment.

Response:

Thank you for pointing out this oversight. We have modified the text as follows:

Page 5, lines 99-102: “We therefore established latent genital infections in mice by intravaginal inoculation with HSV-1 after treatment with Depo Provera, which synchronizes the estrus cycle and increases HSV infection.”

Also figure 1a shows schematic of ocular infection; however part b concerns vaginal infection, so the diagram is only partial. It is thus unclear where AAV were inoculated when looking for vaginal model

efficacy. This bit is thus a little confusing to the first time thru reader, as a person might think the eye is the route for the treatment in the vaginal model. Please make the studies more clearly explained as to what was actually done

Response:

Thank you for this suggestion to help us convey our work clearly. We have modified the schematic and text in figure 1 to clearly depict the vaginal route of infection, and ensured that it is clear in the text.

Lin 107 define BET, it is not a common phrase

Response:

Modified the text as follows:

Page 6, lines 114-117: "The BET (Bromo and Extra-Terminal domain) bromodomain inhibitor JQ1 was reported to reactivate latent HSV in vitro in primary neuronal cultures, and HSV could be detected in the eyes of about one-third of latently-infected mice treated with JQ1"

Line 129 could this be interpreted to support that a large portion of the ganglionic load represents unreactivable genomes (under these conditions?). This has long been speculated and may reflect behaviors of different neuronal subtypes to reactivation stimuli.

Response:

This is a fascinating question, particularly in light of the importance of apparently unreactivable genomes in HIV cure studies. Unfortunately, our study was not designed to address this issue, and we would have little more to offer beyond additional speculation.

Line 277 this is quite similar to VZV reactivations in that a prior event is not a predictor of downstream events. Could this be interpreted to mean that perhaps, the reactivated neuron does not survive the reactivation event?

Response:

This is another interesting question. Unlike VZV (where neuronal loss after reactivation is readily detectable), HSV reactivations are not thought to result in death of the latently infected neuron, largely based on the normal number and distribution of neurons in ganglia from autopsies of people with decades-long symptomatic HSV infection. We have routinely performed histologic examination of ganglia from our experimental animals, and likewise see no evidence of neuronal loss after reactivation. However, we think this issue is somewhat peripheral to the main emphasis of our paper, and would prefer not to comment without performing a dedicated and properly designed study.

Line 337 not quite clear how and where AAV were introduced for vaginal infection and blocking of shedding studies

Response:

We have reviewed the figures and modified as needed to clarify the route of AAV administration in each.

Reviewer #2 (Remarks to the Author):

The authors developed an approach against HSV infection, based on gene editing using HSV-specific meganucleases delivered by adeno-associated virus (AAV) vectors, which is composed of two anti-HSV-1 meganucleases delivered by a combination of AAV9, AAV-Dj/8, and AAV-Rh10 and showed elimination of latent HSV DNA in both ocular and vaginal mouse models of HSV infection. My one of my main concerns is its novelty.

I compared this manuscript with the Nature Communication paper published by the same group in 2020.

The claims of novelty:

1. High efficiency using triple AAV (9, Dj/9, Rh10) infection. But this triple AAV (1+8+Rh10) strategy was used in the previous paper and achieved similar efficiency to clear HSV genomes in SCG and TG peripheral neurons. In addition, the authors used a much higher total AAV dose (3×10^{12} vg/mouse) in this study, which induced toxicity in mice. If the lower dose (1.8×10^{12} vg/mouse) was used (Figure 4a-c), the HSV genome clearing efficiency was actually lower than that achieved in their previous study.

Response:

While we agree that increasing overall genome clearing efficiency is an important goal, we hope to emphasize that the novelty of the paper was largely in 1) extension of gene editing beyond orofacial disease in the previous manuscript to genital infection here; in fact, the genome clearing efficiency in genital disease is the highest we have observed to date, at >97%, 2) establishment of the first reproducible peripheral HSV shedding model in mice, and 3) demonstration that reduction in ganglionic viral load leads to a corresponding reduction in peripheral viral shedding. To address concerns regarding novelty, in the revised paper we now add a series of experiments supporting two new points of emphasis; specifically that 4) a simplified single-AAV/single-meganuclease therapeutic regimen can achieve efficient reduction of latent ganglionic HSV, and 5) toxicity can be greatly ameliorated or even eliminated through dose reduction and tissue restriction of meganuclease expression.

With regard to the specific levels of gene editing after ocular infection, in the previous study with 3 AAVs (1+8+Rh10) delivering two meganucleases, the dose used was 2×10^{12} vg/mouse, and a reduction of 92% in SCG and 54.8% in TG was observed. In Fig 4a-c of the new paper, a different triple AAV regimen (9, Dj/8, Rh10), at a slightly lower dose of 1.8×10^{12} vg/mouse, was used to deliver the same two meganucleases, yielding a similar reduction of 83.3-86.4% in SCG and 48.3-56.1% in TG. Given experimental variability, we do not consider these results meaningfully different. Importantly, in the revised manuscript (Fig 6a-d), we show that a single serotype AAV9 delivering a dual cutting meganuclease m4 (i.e., the simplified regimen) administered at an even lower dose of 1×10^{12} vg/mouse can lead to a similar decrease in ganglionic viral loads (89.6% in SCG and 69% in TG).

2. Established HSV reactivation/shedding model with QJ-1 drug induction. But this model is highly variable. With such high variations, authors didn't do a sample size calculation to determine the animal number for each expt so for some key expts the result were not statistically significant (Figure 2f, 2l, 3j-l, 4g).

Response:

We agree with the reviewer that induction of HSV shedding in mice using JQ1 is variable. We typically use the term “stochastic” for our observations, in the sense of having a random probability distribution or pattern that may be analyzed statistically but may not be predicted precisely. Typically, 30-40% of mice in a given cohort will shed HSV after a given administration of JQ1. But mice that shed HSV after a first administration often do not shed if JQ1 is given at a later time, but may after yet a third JQ1 treatment, and so on. Conversely, animals that do not shed during the first treatment with JQ1 often shed after the second or third administration (or both). Similarly, shedding typically show strong laterality (i.e. shedding is usually limited to one eye), yet the side of shedding after one JQ1 administration does not predict the side of later shedding episodes. While this stochasticity complicates experimentation, we would argue that it also recapitulates the experience in human HSV infections, where shedding occurs episodically, at irregular intervals, and at varying anatomic locations (Johnston et al 2014).

Regarding power calculations, we have performed them based on our experience, and find that the predicted minimum group sizes are very close to the maximum cohort sizes that we can practically do in our lab (which is among the heaviest users of the ABSL2 mouse facility at Fred Hutch) due to limitations in vivarium capacity for virally-infected animals and lab personnel. We fully recognize and discuss in our paper that several individual shedding experiments showed p values >0.05, and this was the impetus for the meta-analysis that we present on pages 10-11, lines 216-243, which shows strong statistical significance of our findings.

Minor issues:

Figure 3g missing

Response:

We thank the reviewer for noticing this, and have reinserted the missing panel.

Figure 2j difficult to see the individual mouse curve without coloring.

Response:

We thank the reviewer this suggestion, and the curves are now colored.

Reviewer #3 (Remarks to the Author):

Keith Jerome's lab presents an interesting follow-up to a gene editing approach to “curing” HSV-1 latent virus from the trigeminal and superior sacral ganglia following eye inoculation and the dorsal root ganglia following intravaginal inoculation. Overall the authors eliminated 97% of “latent” HSV DNA in ganglia. This reduction translated into partial reductions in reactivated virus following treatment with the BET bromodomain inhibitor JQ1 to induce HSV-1 reactivation from latency. The interference with virus reactivation differed substantially in treatment of different ganglia; TG being the least controlled. This strategy might be translated to patients but issues remain regarding methods of delivery, AAV transduction efficiency and vector toxicity.

It is not clear why AAV is being developed and not HSV that will naturally enter the same neuronal cell types without toxicity and one HSV vector can easily be used to express all three nuclease transgenes. Perhaps the authors would like to discuss.

Response:

We agree that HSV itself might be an interesting vector for targeting neurons, and in fact recently deposited a preprint sharing our early results using HSV as the vector for a gene editing/gene drive strategy against HSV (<https://www.biorxiv.org/content/10.1101/2023.12.07.570711>). While the preprint clearly shows that gene drive can occur in mice when wild-type and gene-drive HSV are administered concurrently, critical questions remain as to the efficiency with which HSV vectors can superinfect latently infected neurons, as do important questions about the duration of transgene expression from HSV, and the degree to which HSV vectors will need to be attenuated to ensure safety while maintaining efficient transgene expression.

In contrast, AAV vectors are well understood, have been administered to thousands of participants in human clinical trials, and form the basis of several FDA-approved therapeutics. The results in our paper show their power to reduce ganglionic HSV load and peripheral shedding, as well as strategies to ensure their safe use. Thus, while we continue to evaluate HSV as a vector, at this point we feel that AAV is the more promising approach.

Comments:

1. It would be of significant interest to know the efficiency of uptake of the AAV vectors into sensory neurons and to what extent AAV genomes overlap with latent HSV. In their previous publication, scRNA analysis was used to address this issue. Why wasn't this type of study repeated since one goal here was to enhance transduction efficiency. Since TG treated animals show inefficient blocking of HSV reactivation, is this due to a lack of co-localization of AAV and HSV in neurons particularly since multiple AAV vectors are in use.

Response:

We agree that maximizing the efficacy of anti-HSV gene editing is an important issue, and the topic of continued experimentation in our group. Numerous potential explanations exist for the HSV genomes that remain after AAV/meganuclease administration. As the reviewer points out, we have previously evaluated the distribution of AAV vectors and HSV genomes in treated animals using 10x scRNA analysis, and find that AAV is broadly distributed throughout ganglionic neurons, with substantial co-localization in HSV-infected neurons. At the same time, the results strongly suggest that the 10x approach does not have sufficient sensitivity to definitively answer the co-localization issue. AAV transcripts are detected in a substantially smaller percentage of infected cells than the corresponding reduction of HSV load would suggest, which we interpret to mean that gene editing may occur even in cells with insufficient transgene expression to be detected by 10x. Additionally, it is well known that only a subset of HSV-infected neurons expresses LAT at any given time, and thus could be detected by 10x. Currently, we favor the hypothesis that gene editing measured 1-2 months after AAV/meganuclease administration, as we have done here, underestimates the final "plateau" of absolute efficacy, which may come much later due to the durable nature of AAV transgene expression in neurons. This interpretation is supported by our meta-analysis of our data (pages 10-11, lines 216-243), which showed that the probability of viral shedding significantly decreased with the duration of meganuclease therapy ($p < 0.001$). We are currently performing long-term efficacy and safety studies that should shed additional light on these issues, but their duration (approximately 2 years) precludes inclusion in these revisions.

Three different meganuclease expressing vectors are presumably needed to create multiple single stranded breaks in the HSV genome possibly to prevent possible HSV DNA repair. The nuclease

cDNA is likely about 3kb that necessitates three different vectors, an expensive and complex strategy for manufacture?

Response:

This is not correct. A single meganuclease induces a DNA double strand break at its target site (in contrast to ZFNs or TALENs that need to be paired to generate DNA double strand breaks, as each binding domain is fused to a DNA-cleavage domain of the restriction endonuclease FokI which must dimerize to cleave DNA). The purpose of pairing the m5 and m8 meganucleases in the original experiment was to induce two DNA double strand breaks within the latent HSV episome, which we demonstrated in our 2020 Nature Communications paper favors degradation of the HSV episome, rather than repair of the cleavage site, which is more common after inducing only one DNA double strand break. The coding sequence for a meganuclease is approximately 1 kb, which when combined with promoter and enhancer elements in self-complementary AAV construct, yields a transgene size of almost exactly the ideal length for optimal packaging efficiency into AAV particles. We consistently achieve high viral titer and purity of our AAV stocks, prepared in-house using standard laboratory procedures. To clarify the structure of our AAVs, we have revised their description in the methods section, and added a supplemental figure of our AAV plasmids used to produce the AAV delivery vectors:

Page 20, lines 460-462: “AAV production and titering. AAV vector plasmids pscAAV-CBh-m5, pscAAV-CBh-m8, pscAAV-CBh-m4, pscAAV-CBh -m4i and pscAAV-E/CamKII-m4 were used to generate the AAV stocks in this study (Fig. S12).”

On the other hand, we appreciate the general concern regarding the complexity of the treatment regimen in our original submission, which consisted of six different constructs (3 AAV serotypes x 2 meganucleases). We therefore performed a series of experiments designed to greatly simplify the regimen, and in the revised manuscript show strong anti-HSV activity with a regimen consisting of a single AAV serotype (AAV9) delivering a single meganuclease, m4, which targets a duplicated site within the HSV genome, and thus can create two DNA double strand breaks in the HSV episome by itself. We are grateful to the reviewer for prompting these experiments, which have resulted in a regimen much easier to manufacture and which will likely more easily pass regulatory review as we move toward clinical application. These experiments are presented in the revised manuscript on pages 11-13, lines 245-273, in the section entitled, “Simplification of the AAV-meganuclease regimen.”

Is the assumption that the different AAV vectors infect the same or different neuronal cell populations?

Response:

The 10x scRNAseq data in our published work (Aubert et al 2020) suggested that different AAV serotypes transduced neuronal populations within the TG and SCG with different efficiency, which we believe accounted for the superiority of triple-AAV regimens in those studies. In subsequent optimization studies evaluating addition serotypes, presented in the revised manuscript, we find that AAV9 alone can achieve overall gene editing results equivalent or even superior to our best triple-AAV regimens, thus allowing the regimen simplification discussed above.

I suspect that the distribution of AAV differs substantially in different test applications which accounts

for why some animals are unaffected by the AAV treatment. Delivery remains an important high bar for effective therapy.

Response:

We agree that delivery is an important factor for effective therapy, and as noted above we have performed extensive optimization that has allowed us to achieve efficient gene editing of HSV using the single AAV serotype AAV9 and a single meganuclease m4, which targets a duplicated sequence in the HSV genome. We would point out that in fact while the absolute degree of latent HSV reduction varies between animals, we have not observed any truly unaffected animals, in that all show at least some degree of gene editing.

2. Two anti-HSV-1 meganucleases were delivered using a combination of AAV vectors (AAV9, AAV-Dj/8 and AAV-Rh10) to attack resident HSV DNA.

a- What are the specific targets? Earlier work targeted VP5, polymerase and ICP0.

Response:

We have modified the text to clarify the targets for each of our meganucleases:

Page 5, lines 91-92: "...the anti-HSV1 meganuclease m5, which cleaves a sequence in the UL19 gene coding for the major capsid protein VP5,"

Page 5, lines 104-105: "... along with m8, which targets a sequence in the UL30 gene coding for the catalytic subunit of the viral DNA polymerase."

Page 12, lines 250-251: "...the dual cutting meganuclease m4, which recognizes a sequence in the duplicated gene ICP0 in the HSV-1 genome..."

b- There are no data on meganuclease expression in vivo or HSV DNA mutations.

Response:

We have generated data showing meganuclease expression in vivo (Supplemental figure 11), and analyzed HSV DNA mutations by T7 analysis (presented in Supplemental Fig. 1 and 2). We have also added results of T7 analysis for the m5 target site in the experiment presented in Fig.1b (panel e), Fig.2a (panel d) and Fig.2g (panel j). The text was modified as follows:

Pages 13-14, lines 293-297: "Assessment of m4 expression in neuronal tissues at different times post administration of either AAV9-CBh-m4 or AAV9-E/CamKII-m4 showed that the m4 expression increased over time but was in general slightly lower in tissues from AAV9-E/CamKII-m4-treated mice compared with AAV9-CBh-m4-treated mice (Fig. S11a-b)."

3. Because initial virus shedding from the TG was not affected by AAV expression of nucleases but reduced substantially in later drug inductions, how will this be useful to treatment of human disease. Generally human reactivations are sporadic.

Response:

As discussed above, we consider the stochastic nature of JQ1 induced reactivation to be analogous to the sporadic reactivations in humans noted by the reviewer. The limited effect of

AAV/meganuclease treatment on initial virus shedding was observed only with the lower dose of AAV/meganuclease in Fig. 3. However, an effect on shedding during the initial reactivation period was observed with higher doses (Figs. 2, 4, 6). We cannot rule out that the limited effect in Fig. 3 simply represents experimental noise due to stochasticity. However, it is tempting to speculate that it may relate to ongoing gene editing over time between the reactivations (as we discussed above), which would be more easily detected at lower AAV/meganuclease doses. As mentioned previously, we have a long-term experiment ongoing (expected completion in about 18 months) to evaluate this issue. If gene editing efficacy improves over time, human recipients of our therapy could expect to see continued improvement over time post-therapy.

4. The reduction of HSV genomes in the TG would seem to be most important given the clinical symptoms of herpes keratitis. Fig 1 shows a reduction of 61% and 42% reduction in latent virus but no statistically significant reduction in TG reactivation (Fig. 2). What accounts for this difference. Is there evidence of inflammation of the cornea?

Response:

The lack of statistically significant reduction in Fig. 2 reactivation is reviewed in the discussion. We attribute this to the episodic and stochastic nature of the shedding after reactivation, and the challenge that this constitutes to adequately powering individual experiments. This is the reason that we performed the meta-analysis of the combined data from all our experiments, which confirms that the reduction in HSV shedding from AAV/meganuclease-treated animals compared with controls is indeed statistically significant.

5. Among controls and after JQ1 treatment, how many neurons show active HSV transcription of a standard lytic gene such as glycoprotein D compared with attacked viral gene.

Response:

This question would be extremely technically challenging to answer, since during viral reactivation it is believed that only a very small number of neurons reactivate, preventing determination of the proportion of neurons showing transcription of a “standard lytic” gene (gD) versus the “attacked gene” (UL19 or UL30). Furthermore, as we discussed extensively in our 2020 Nature Communications paper, the power of our approach lies not in disruption and mutagenesis of the targeted viral gene (as predominates after inducing only on DNA double strand break in the HSV episome), but instead elimination of the entire targeted episome, which occurs preferentially when two DNA double strand breaks are induced within the episome,

6. Do the authors know how many neurons show virus reactivation. In natural infections, this number could be very low which speaks to the importance of efficient co-localization of AAV and HSV in neurons. Drug treatment may activate many neurons to release HSV but in human infections this could be limited and hit or miss as to whether AAV has inactivated HSV in reactivating neurons.

Response:

As noted in the previous response, this would be challenging to address experimentally. We agree that the number of neurons reactivating in humans is likely low, as evidenced by the observations that shedding is episodic and stochastic, and can occur at distinct anatomical locations during different shedding episodes. As discussed previously, we observe these same characteristics (episodic, stochastic, and anatomically localized shedding) after JQ1

administration in mice, suggesting that the number of neurons is likely similarly low. We have noted these findings in the manuscript:

Pages 6-7, lines 132-135: “shedding was typically unilateral (only detected in one eye), despite the initial inoculation being to both eyes (unilateral shedding was observed in 33/37 (89%) of events). The side of shedding in one episode was not predictive of the side of future shedding events (Table S1).”

7. Since the reduction in HSV virus load differed substantially between SCG and TG, does this reflect AAV transduction efficiency of HSV containing neurons?

Response:

The superior efficiency of anti-HSV gene editing in SCG (typically ~90% reduction in latent HSV genomes) compared to TG (typically 50-60% reduction) is a remarkably consistent finding, which we have observed over literally dozens of experiments since starting this project. As mentioned in our response to the reviewer’s point #4, we evaluated potential explanations for this in our Nature Communication publication of 2020, where we showed that AAV transduction of HSV containing neurons was more effective in SCG than TG. One potential explanation was that AAV transduction might be intrinsically more efficient in autonomic ganglia (like the SCG) than in sensory ganglia (like the TG). Thus, a critical question addressed in our current manuscript was whether HSV reductions in DRG (also sensory ganglia, and the critical ganglia for genital HSV infections) would be limited to 50-60% like TG. We were of course very pleased to observe that latent HSV was reduced in DRG even more efficiently (up to 97%, see Fig. 1c) than it is in SCG. We currently favor the hypothesis that the relative efficiencies of AAV transduction in the various ganglia are driven mainly by the relative permeability of the blood/ganglionic barrier for each ganglionic type. We currently have a series of experiments underway to address this issue, including comparison of transduction efficiencies in each ganglionic type after intravenous vs. intrathecal administration. In any event, the main point we wish to stress in the current manuscript is that DRG, the critical ganglionic type in genital HSV infections, is readily accessed by AAV and fully amenable to gene editing of latent HSV genomes. We have added the following paragraph to the discussion to address these points:

Pages 17-18, lines 375-385: “A second important finding in our studies is that AAV-delivered meganucleases can readily enter neurons and edit HSV within DRG, the site of HSV latency in genital disease. We have consistently observed superior HSV viral load reduction in SCG (which are autonomic ganglia; typically ~90% reduction) compared to TG (sensory ganglia; typically ~50-60% reduction). This raised the possibility that DRG (also sensory ganglia) might also show similar modest rates of genome reduction. However, gene editing of latent HSV genomes proved to be highly efficient within DRG (97% reduction), suggesting that the differing efficiencies are not intrinsic to the type of ganglion (sensory vs. autonomic). We currently favor the hypothesis that the relative efficiencies of AAV transduction in the various ganglia are driven mainly by the relative permeability of the blood/ganglionic barrier for each ganglionic type, and have a series of experiments underway to address this issue.”

What are the kinetics of AAV transgene expression. It might be expected to come on slowly and peak after weeks. How might this affect efficiency of KD of HSV genomes?

Response:

The reviewer is exactly correct in their prediction. In our publication Dang et al 2017, we showed that while the levels of AAV genomes in the ganglia peak quickly (3 days post administration), transgene expression slowly increased over the 28 days of the study, and appeared to be still increasing at study termination. Thus, our typical analysis, performed after approximately 1 month after AAV/meganuclease (for reasons of experimental convenience; latency establishment takes one month, as does full analysis of tissue obtained after therapy), appears to underestimate the ultimate efficacy of latent HSV depletion. This is supported by our meta-analysis showing increasing HSV depletion over time and is discussed above.

The data suggest that the treatment improves over time due AAV based gene expression however in applications in which the timing of AAV administration will be difficult to assess how might this be dealt with? The authors need to address this in the discussion.

Response:

We may not fully understand the reviewer's question, as we cannot imagine a circumstance, experimental or clinical, in which the time of AAV administration would not be known. However, the reviewer is correct that the efficacy of treatment improves over time, and we would expect this to manifest in improved clinical response over time as well.

8. The HSV inoculates are at one dose. There needs to be a dosing experiment to assess HSV DNA destruction efficiency. How does the distribution and amount of HSV DNA correlate with the number of genome copies per neuron and the impact on AAV-mediated genome reduction. The number of latent genome copies per cell is known to vary considerably from a few to thousands of copies. Will high copy DNA neurons be resistant to treatment?

Response:

Our HSV inoculate levels were chosen based on the literature and our own empiric experience to give the best balance between the morbidity and mortality observed during the acute phase of HSV infection, and the level of ganglionic latency ultimately achieved. We typically achieve ganglionic latent HSV loads that are similar to those observed in humans, with acceptable morbidity and mortality. It would not be possible to determine whether high copy DNA neurons would be resistant to treatment, as it is not possible to measure ganglionic (or individual neuronal) viral loads prior to treatment. On the cohort level, there is clearly a distribution of ganglionic viral loads in control animals; a similar distribution is observed in treated mice, but at about 1 log lower levels. The most straightforward interpretation is that HSV DNA, at any level, is susceptible to meganuclease attack and degradation, but it is likely that the degree of absolute reduction would vary between individual animals, were we able to measure it.

9. Using only HSV-1 as a model system in vaginal infections does not adequately reflect the fact that the majority of vaginal infections are HSV-2 which may be more difficult to knock down. Do the selected meganucleases destroy HSV-2 DNA? Guinea pigs would be a more adequate model system since natural recurrence occurs in these animals.

Response:

As noted in the response to reviewer 1, the HSV-1 target sequences of the meganucleases used in this study are not well conserved in HSV-2. We performed the vaginal infection experiments to make the point that there appears to be no barrier to treating genital vs. orofacial disease; that is, the latently infected neurons in both sites are readily accessible to AAV vectors as well as meganuclease-mediated editing of latent HSV genomes. We would point out, however, that

successful treatment of genital HSV-1 infection is not a trivial result – over half of new cases of genital herpes in the US are now due to HSV-1. Nevertheless, we agree with the reviewer that genital HSV-2 shedding is an important target. We have rationally engineered a series of anti-HSV-2 meganucleases that are currently being tested in vitro and in vivo. However, that work is not yet mature enough for publication, and will be the subject of a future report. We also agree regarding the advantages of the guinea pig model, particularly the spontaneous HSV recurrences, although these must be weighed against the substantially greater space requirements and cost of the model. We have performed initial studies in guinea pigs (with HSV-1 for now), and observe clear evidence of gene editing and a statistically significant reduction in recurrence severity. Again, however, the work is not yet mature enough for publication, and we would prefer to present a complete set of guinea pig studies in a future report. As noted in the response to reviewer 1, we have added a paragraph addressing these points to the discussion section:

Page 18, lines 387-395: “One limitation of our work is that the HSV-1 target sequences of the meganucleases used in this study are not well conserved in HSV-2. We therefore performed the vaginal infection experiments using HSV-1, to make the point that there appears to be no barrier to treating genital vs. orofacial disease; that is, the latently infected neurons in both sites are readily accessible to AAV vectors as well as meganuclease-mediated editing of latent HSV genomes. We would point out that successful treatment of genital HSV-1 infection is not a trivial result – over half of new cases of genital herpes in the US are now due to HSV-1²⁷. Nevertheless, we fully recognize the importance of HSV-2 as a target, and current work in our laboratory focuses on development of anti-HSV-2 meganucleases and their testing in mice and other models of infection.”

10. Following vaginal infections, which DRG are harboring HSV?

Response:

After vaginal infection, latent viral genomes are mainly found in the lumbosacral DRG (Richards et al 1981).

11. I assume that AAV will persist in various tissues including sensory neurons. Over time one might expect the meganuclease protein to induce an immune response. Have authors looked for this?

Response:

We agree with the prediction that there may be an immune response to the meganuclease protein, We are currently evaluating T and B cell responses in long-term tolerability/safety studies, especially in the context of the dose reduction and tissue restriction strategies we have added to the revised paper.

12. About a third of individuals have anti-AAV neutralizing antibodies. Will this anti-vector immune activity affect the efficiency of transduction of sensory neurons?

Response:

The reviewer is correct that a substantial fraction of individuals have pre-existing antibodies to AAV, including AAV9. We expect that such antibodies would prevent efficient transduction of neurons after intravenous administration of AAV. We are currently evaluating the relationship of antibody titer to transduction efficiency, and testing the hypothesis that direct intrathecal injection

of AAV may circumvent the inhibitory effect of serum antibody, as has recently been suggested (Gray et al, 2013; Horiuchi et al 2022).

13. Delivery of AAV to sensory could be highly problematic and potentially cytotoxic as observed in this study. How do the authors envision vector delivery in a human application? High dose AAV delivery to ganglia in primates has recently been shown to be highly cytotoxic so safety will be an issue (Hum Gene Ther. 2018 Mar;29(3):285-298). Toxicity was independent of the transgene and virus capsid. The authors indicated that they did not see detectable signs of neuropathy following vector treatment however there was evidence of neurodegeneration. I would imagine that this could lead to neuropathic pain. This potential safety problem needs to be evaluated.

Response:

We agree that safety issues need to be addressed continuously while the work progresses, as we stated in our response to reviewer 1. We are well aware of the issues with high dose AAV as noted by the reviewer, and in fact extensively reviewed the literature (including regarding AAV neurotoxicity (including the paper cited by the reviewer) in a manuscript we published earlier this year in Gene Therapy (<https://doi.org/10.1038/s41434-023-00405-1>). While we cannot go into the same depth here, we would note that the AAV dosage levels likely to lead to hepatotoxicity are well understood, and correlate well on a vector genome/kg basis with the hepatotoxic levels in many animal models including our mice. AAV-associated neurotoxicity has been only more recently appreciated. It appears to be mainly histologic, with no associated deficits or neuropathies reported (with one possible exception) over thousands of human participants in AAV trials. Similarly, our mice show no clinical signs of neuropathy,

Nevertheless, we agree this is a critical issue, and in response to the comments, we extended our safety studies, and in the revised paper demonstrate that undesirable effects on both the liver and ganglia can be essentially eliminated through regimen simplification (to a single AAV serotype and single meganuclease), dose reduction, and tissue restriction of meganuclease expression. These data have been added to the manuscript in two new sub-sections of the results, "Simplification of the AAV-meganuclease regimen" and Tissue restriction of meganuclease expression improves tolerability", found on page 11, line 245, through page 13, line 273.

14. How does AAV therapy compare with ganciclovir treatment, the standard of care regarding reactivated virus load?

Response:

We presume the reviewer meant acyclovir, which is the standard first-line treatment for HSV (ganciclovir is predominantly an anti-CMV drug; while it has anti-HSV activity similar to acyclovir, it is substantially more toxic). Acyclovir and derivatives can reduce the severity of primary and recurrent disease, as well as the frequency of virus shedding. However, symptoms are typically only shorted by 1-2 days, and shedding can still occur while on suppressive acyclovir therapy, with transmission risk estimated to be reduced by about 50%. Importantly, acyclovir and derivatives do not affect latent virus or reduce ganglionic viral load, and consequently, once antiviral treatment is stopped viral shedding and symptoms typically recur rapidly. In contrast, AAV/meganuclease therapy need be administered only once, directly attacks latent HSV, and offers the potential of durable benefit. As discussed above, even when evaluating only one month after therapy (which likely underestimates the ultimate efficacy), we see suppression of viral shedding similar to that achieved on suppressive acyclovir therapy.

Reviewer #1 (Remarks to the Author):

Aubert et al extend previous studies evaluating the mouse model of ocular and vaginal infection and the ability to reduce the genome copy number of the latent virus load, particularly as it relates to the ability of virus to be experimentally reactivated by the Bromodomain inhibitor JQ-1. Since the original submission, numerous additional experiments have been done and the paper largely rewritten. There are some exciting new studies, including a meta-analysis of all the reactivations to apply a statistical assessment of the reduction in reactivation frequency, and the development of cell type specific promoters as an improved safety component. This would be generally a paper of wide interest. I will be honest I do have to say that this version is quite a dense read, not as clearly written as the original; There are several additional studies in the revised version and a lot of supplemental studies that do not always seem consistent. The rebuttal is comprehensive and well put and they have done an excellent job at trying to satisfy the issues brought up by the reviewers. I liked to see the additional data on reactivation (JQ versus Heat Shock for example). They also expand the safety analyses, and show a minimalization using a simplified delivery strategy with a single dual cutting meganuclease (I do not know why this strategy was not adopted from the start- it makes much more sense. The authors should indicate this might be a strategy that should be applied elsewhere. In addition, there is a nice new and important study that assessed the use of specific neuron promoters for meganuclease expression.

comments

1. Most of their controls are simply untreated (no AAV). I am sure they have determined that AAV itself does not induce any clearance of HSV-1 such as by induction of innate immunity? If so, please remind the reader in a short line stating control untreated versus control vector only. Indeed, the inactive meganuclease studies in figure 6 with M4 confirm this statement that AAV (at least regarding AAV9) that control does not reduce ganglionic loads
2. In supp fig1 they show high transduction by RO of DJ8 delivery and a corresponding high mutation rate, but this seems in contrast to fig 1 c and d where it seems rather different. Is this worth at least a comment on variable studies? It seems to reduce SCG levels effectively in one panel but not in 1 C+ D
3. Why do controls (no AAV) have detectable AAV genomes in them, some of them showing quite high levels, in supplemental fig 2?
4. Statements are often rather general: text indicates there is a global "especially good efficacy across both SCG and TG" but for most TG in supp fig 1 there was no significance. I would not call the editing "robust" (line 95, line 109) since there is still only a small fraction that have mutations
5. Several figures seem not to be referred to in the text (S2d-G)
6. Why does Figure 2 I and J have a lot of lines overlapping each datapoint for the AAV/MN?
7. For some figures, points are very small and maybe hard for the reader to see clearly (SFig3)
8. For Sfig4 reactivations are labelled 32, 39 and 46, but it is a bit confusing considering the data figures referring to 7 and 14 days. I suggest adding a second X label of days post infection. Also, label should be "days after first reactivation stimulus" for supp 4b-e
9. Line 137 could this mean the reestablishment of equivalent latent load after reactivation?
10. Line 153 should be 2h not 2g (I think)
11. I still think the vaginal infection data and the poor reactivation induction is a little detracting from the excellent ocular studies, but that is perhaps just this reviewer. They made a strong argument for keeping it in
12. The newly added meta-analyses were a nice addition, and supportive of their total studies and conclusions that genomes can be clinically reduced to reduce reactivation frequencies
13. Line 236 define GLMM
14. New studies show the use of more neuron specific promoters for the expression of the nuclease and the reduction of toxicity. These greatly add to the paper and the importance of safety
15. Since the original submission of this paper, there have been some advancement in the application of gene editing strategies to remove HSV in humans. It might well be appropriate to mention this in

the discussion, as the authors are clearly aiming for this goal.

Reviewer #2 (Remarks to the Author):

all my concerns have been satisfactorily addressed

Reviewer #3 (Remarks to the Author):

After careful reading of the revised manuscript and responses to reviewers I've concluded that the paper makes an interesting contribution to the use of a gene editing AAV vector to reduce the load of latent HSV in TG and SCG. An important advance was the development of a vector expressing a single meganuclease (m4) to cleave latent HSV genomes in vivo through recognition of ICP0 sequences that in turn create several double stranded cuts in the HSV episomal circular genome. Previously this approach required the use of 3 distinct AAV gene editing vectors. This innovation reduced vector treatment dose. M4 expression was limited to neurons using a neuronal promoter(CamKII). While safety was further improved, this vector was less effective. I'm not sure if this works in the absence of HSV reactivation but there was a substantial percentage reduction in HSV genome load. Whether this would be meaningful in terms of re-establishment of local infection and transmission to other hosts is an open question. Referring to percentages rather than virus numbers could be misleading and it is not clear how many particles are needed to establish local replication.

RESPONSE TO REVIEWER COMMENTS

We thank the reviewers for their thoughtful review and helpful suggestions to improve the manuscript. We have made the necessary modifications to address their comments, as detailed below, and incorporated their suggestions into the revised manuscript.

Our point-by-point responses to the reviewers' comments are as follows:

Reviewer #1 (Remarks to the Author):

Aubert et al extend previous studies evaluating the mouse model of ocular and vaginal infection and the ability to reduce the genome copy number of the latent virus load, particularly as it relates to the ability of virus to be experimentally reactivated by the Bromodomain inhibitor JQ-1. Since the original submission, numerous additional experiments have been done and the paper largely rewritten. There are some exciting new studies, including a meta-analysis of all the reactivations to apply a statistical assessment of the reduction in reactivation frequency, and the development of cell type specific promoters as an improved safety component. This would be generally a paper of wide interest.

I will be honest I do have to say that this version is quite a dense read, not as clearly written as the original; There are several additional studies in the revised version and a lot of supplemental studies that do not always seem consistent. The rebuttal is comprehensive and well put and they have done an excellent job at trying to satisfy the issues brought up by the reviewers. I liked to see the additional data on reactivation (JQ versus Heat Shock for example). They also expand the safety analyses, and show a minimalization using a simplified delivery strategy with a single dual cutting meganuclease (I do not know why this strategy was not adopted from the start- it makes much more sense. The authors should indicate this might be a strategy that should be applied elsewhere. In addition, there is a nice new and important study that assessed the use of specific neuron promoters for meganuclease expression.

Response: *We thank the reviewer for these comments. Regarding the suggestion that we should indicated that the dual cutting strategy should be applied elsewhere, in **Line 411-414**, we have added the following: "We are especially interested in targeting duplicated regions of HSV-2, which our results here suggest will allow effective single-nuclease therapy, and we would suggest that duplicated or repeated sites should be considered for gene editing efforts targeting other viruses."*

1. Most of their controls are simply untreated (no AAV). I am sure they have determined that AAV itself does not induce any clearance of HSV-1 such as by induction of innate immunity? If so, please remind the reader in a short line stating control untreated versus control vector only. Indeed, the inactive meganuclease studies in figure 6 with M4 confirm this statement that AAV (at least regarding AAV9) that control does not reduce ganglionic loads

Response: *The reviewer is correct, the inactive meganuclease studies in figure 6, as well as previous work with irrelevant nucleases or other transgenes, clearly demonstrate that the anti-HSV effect is dependent on the presence of the HSV-targeting meganuclease. To clarify this point, in **Line 285-288**, we have added the following: "These data demonstrate that our simplified regimen can substantially reduce ganglionic viral loads, with an associated decrease in virus shedding after reactivation, and that these effects are dependent on an active enzyme and not on AAV itself."*

2. In supp fig1 they show high transduction by RO of DJ8 delivery and a corresponding high mutation rate, but this seems in contrast to fig 1 c and d where it seems rather different. Is this worth at least a comment on variable studies? It seems to reduce SCG levels effectively in one panel but not in 1 C+ D

Response: We assume that in the second part of the question, the reviewer is referring to Supplemental Fig.1, panel c and d, since Fig 1 in the main text does not address the activity of Dj/8. In Supplemental Fig.1, panel b shows the reduction of ganglionic HSV after therapy, while in contrast, panels c and d show the AAV loads and percent of mutation within residual HSV post therapy, respectively. We have ensured that the axis labels and figure legend are clear on the differences between the panels.

3. Why do controls (no AAV) have detectable AAV genomes in them, some of them showing quite high levels, in supplemental fig 2?

Response: We have a statement in the Material and Methods section to explain this observation: **line 503-506**, “In some control ganglia, sporadic samples showed positivity for AAV genomes, although the levels were typically >2-3 logs lower than in ganglia from treated mice having received AAV. We attribute this to low-level contamination of occasional tissue samples.”

4. Statements are often rather general: text indicates there is a global “especially good efficacy across both SCG and TG” but for most TG in supp fig 1 there was no significance. I would not call the editing “robust” (line 95, line 109) since there is still only a small fraction that have mutations

Response: As we demonstrated in our 2020 Nature Communications paper and confirmed here, gene editing of episomal targets such as HSV can result in either mutations at the target site, or elimination of the targeted episome, and cleavage of the episome at two sites favors the latter (elimination). Thus, “robust HSV gene editing” refers to the sum of these (typically around 90% elimination of episomes in TG, plus the 5-20% mutagenesis detected within the residual genomes). To clarify this point, we have modified **line 94-97** to read, “In agreement with our previous results, combinations of AAV serotypes led to robust HSV gene editing, with the triple combination of AAV9, AAV-Dj/8, and AAV-Rh10 showing especially strong reductions in HSV loads and mutagenesis of residual HSV across both SCG and TG (Fig. S2b-g)”, and **line 109-114** to read, “This compared favorably with the orofacial infection group treated in parallel, in which (in agreement with our previous studies) we observed robust gene editing with significant reductions of ganglionic HSV loads of 89% in SCG and 61% in TG (Fig. 1d-e).”

5. Several figures seem not to be referred to in the text (S2d-G)

Response: Line 94-97. The text was modified as follows: “In agreement with our previous results, combinations of AAV serotypes led to robust HSV gene editing, with the triple combination of AAV9, AAV-Dj/8, and AAV-Rh10 showing especially strong reductions in HSV loads and mutagenesis of residual HSV across both SCG and TG (Fig. S2b-g)”

6. Why does Figure 2 I and J have a lot of lines overlapping each datapoint for the AAV/MN?

Response: In Figure 2i and j for the AAV/MN group, many of the HSV load values are very similar between animals in the cohorts. To make the data points more visually distinct despite their similarity, we decreased the line thickness and used split axes.

7. For some figures, points are very small and maybe hard for the reader to see clearly (SFig3)

Response: *The size of the graphs and the font in Supplemental Fig. 3 were increased to make it easier for the reader.*

8. For Sfig4 reactivations are labelled 32, 39 and 46, but it is a bit confusing considering the data figures referring to 7 and 14 days. I suggest adding a second X label of days post infection. Also, label should be “days after first reactivation stimulus” for supp 4b-e

Response: *To make it less confusing, the figure legend was modified to read as follows: “HSV titers in eye swabs collected daily for 3 days after the 1st (day 32 p.i. in **a** and day 0 in **b-e**), 2nd (day 39 p.i. in **a** and day 7 on graph **b-e**) or 3rd (day 46 p.i. in **a** and day 14 in **b-e**) IP injection of either vehicle (black arrows) or JQ1 (red arrows, 50 mg/kg)”.*

9. Line 137 could this mean the reestablishment of equivalent latent load after reactivation?

Response: *In this experiment, none of the animals received AAV/MN treatment. We compared the ganglionic viral loads in latently infected mice administered 1 to 7 JQ1 injections to those from control mice latently infected with HSV which did not receive any reactivation stimulus. The ganglia from all the mice were collected at the end of the experiment. Therefore, JQ1 reactivation did not result in an increase or decrease of the viral loads.*

10. Line 153 should be 2h not 2g (I think)

Response: *The reviewer is correct. The text was modified accordingly.*

11. I still think the vaginal infection data and the poor reactivation induction is a little detracting from the excellent ocular studies, but that is perhaps just this reviewer. They made a strong argument for keeping it in
and

12. The newly added meta-analyses were a nice addition, and supportive of their total studies and conclusions that genomes can be clinically reduced to reduce reactivation frequencies

Response: *We thank the reviewer for these comments and agree.*

13. Line 236 define GLMM

Response: *We have defined GLMM in the text **line 242-243**. It now reads: “...(OR = 0.41, p = 0.010, by generalized linear mixed models, GLMM).”*

14. New studies show the use of more neuron specific promoters for the expression of the nuclease and the reduction of toxicity. These greatly add to the paper and the importance of safety

Response: *We agree with the reviewer.*

15. Since the original submission of this paper, there have been some advancement in the application of gene editing strategies to remove HSV in humans. It might well be appropriate to mention this in the discussion, as the authors are clearly aiming for this goal.

Response: *We have added the following in the discussion **line 441-446**: “This is supported by the recent report of the use of gene editing strategy to remove HSV from human in an investigator-initiated, open-label, single-arm, non-randomized interventional trial in 3 patients with severe refractory herpetic stromal keratitis (HSK). In this study, no off-target cleavages or systemic adverse events were detected in the 18 months follow-up, while preventing viral relapse (PMID: 37658603).”*

Reviewer #2 (Remarks to the Author):

all my concerns have been satisfactorily addressed

Reviewer #3 (Remarks to the Author):

After careful reading of the revised manuscript and responses to reviewers I've concluded that the paper makes an interesting contribution to the use of a gene editing AAV vector to reduce the load of latent HSV in TG and SCG. An important advance was the development of a vector expressing a single meganuclease (m4) to cleave latent HSV genomes in vivo through recognition of ICP0 sequences that in turn create several double stranded cuts in the HSV episomal circular genome. Previously this approach required the use of 3 distinct AAV gene editing vectors. This innovation reduced vector treatment dose. M4 expression was limited to neurons using a neuronal promoter(CamKII). While safety was further improved, this vector was less effective.

I'm not sure if this works in the absence of HSV reactivation but there was a substantial percentage reduction in HSV genome load. Whether this would be meaningful in terms of re-establishment of local infection and transmission to other hosts is an open question. Referring to percentages rather than virus numbers could be misleading and it is not clear how many particles are needed to establish local replication.

Response: *We agree with the reviewer that establishment of local infection (lesions) and transmission to new hosts are important and outstanding questions. HSV reactivation in mice using JQ1 is generally not associated with lesions, but we are currently evaluating lesion frequency and severity in treated vs. control guinea pigs, and we hope this will be the focus of a future report. We address the issue of transmission in lines 353-363 of the discussion. While there are reasons for cautious optimism that AAV/meganuclease therapy may reduce the risk of transmission, this remains to be formally tested.*

We considered expressing viral load reduction in logs rather than percent (in the DRG in particular, the reduction approaches two logs), but felt that percentage reduction is more accessible in the current context and is justified given the precision of our digital PCR readout. Absolute virus numbers are accessible to the reader from the y-axis (HSV genomes/10⁶ cells) of relevant panels of each figure.

Reviewer #1 (Remarks to the Author):

the authors have addressed all my concerns
signed Paul R Kinchington as reviewer 1